# Chemistry and Pharmacology of Bergenin or Its Derivatives: A Promising Molecule

**DOI:** 10.3390/biom13030403

**Published:** 2023-02-21

**Authors:** Zeca M. Salimo, Michael N. Yakubu, Emanuelle L. da Silva, Anne C. G. de Almeida, Yury O. Chaves, Emmanoel V. Costa, Felipe M. A. da Silva, Josean F. Tavares, Wuelton M. Monteiro, Gisely C. de Melo, Hector H. F. Koolen

**Affiliations:** 1Programa de Pós-graduação em Medicina Tropical, Universidade do Estado do Amazonas, Manaus 69040-000, Brazil; 2Fundação de Medicina Tropical Dr. Heitor Vieira Dourado, Manaus 69040-000, Brazil; 3Instituto Leônidas e Maria Deane, Fundação Oswaldo Cruz, Manaus 69057-070, Brazil; 4Departamento de Química, Universidade Federal do Amazonas, Manaus 69067-005, Brazil; 5Centro de Apoio Multidisciplinar, Universidade Federal do Amazonas, Manaus 69067-005, Brazil; 6Programa de Pós-graduação em Produtos Naturais e Sintéticos Bioativos, Universidade Federal da Paraíba, João Pessoa 58051-900, Brazil; 7Grupo de Pesquisa em Metabolômica e Espectrometria de Massas, Universidade do Estado do Amazonas, Manaus 69065-001, Brazil

**Keywords:** natural products, bergenin, biosynthesis, plants with bioavailability, pharmacological and biological activities

## Abstract

Bergenin is a glycosidic derivative of trihydroxybenzoic acid that was discovered in 1880 by Garreau and Machelart from the rhizomes of the medicinal plant *Bergenia crassifolia* (currently: *Saxifraga crassifolia*—Saxifragaceae), though was later isolated from several other plant sources. Since its first report, it has aroused interest because it has several pharmacological activities, mainly antioxidant and anti-inflammatory. In addition to this, bergenin has shown potential antimalarial, antileishmanial, trypanocidal, antiviral, antibacterial, antifungal, antinociceptive, antiarthritic, antiulcerogenic, antidiabetic/antiobesity, antiarrhythmic, anticancer, hepatoprotective, neuroprotective and cardioprotective activities. Thus, this review aimed to describe the sources of isolation of bergenin and its in vitro and in vivo biological and pharmacological activities. Bergenin is distributed in many plant species (at least 112 species belonging to 34 families). Both its derivatives (natural and semisynthetic) and extracts with phytochemical proof of its highest concentration are well studied, and none of the studies showed cytotoxicity for healthy cells.

## 1. Introduction

The rapid development of the chemistry of natural products has led to the isolation of a variety of secondary metabolites. In particular, bergenin, also known as ardisic acid B, bergenit, bergenitol, cuscutin, peltophorin, and vakerin [1,2]. Its name is derived from where it was isolated, i.e., from the ornamental and medicinal plant *Bergenia crassifolia* L. (currently: *Saxifraga crassifolia* L.—Saxifragaceae), and it was obtained from the rhizomes of this plant [1,2]. Although originally obtained from a small plant distributed in the temperate regions of North-Central Asia, mainly in Russia, the bergenin molecule has been found in several plant species distributed worldwide. To date, bergenin has been obtained from different species of different families (Table 1).

Plants biosynthesize bergenin and other natural compounds as an adaptive mechanism in response to abiotic and biotic stresses, in addition to attracting animals and protecting against ultraviolet radiation [3]. In recent years, bergenin has received increasing attention due to its presence in food and medicinal plants, including the Amazonian plant “uchi” (*Endopleura uchi*). Its fruit is used as a food item and as a medicine and is consumed raw or as a juice, in ice cream or popsicles, and the oil produced from its seeds can be used in foods and for the treatment of sinusitis in children and constipation in adults. Its seeds are used in making handicrafts, smoking meats, and as amulets [4]. Studies with uchi fruit pulp have indicated a rich nutritional composition (fatty acids, fiber, steroids, mineral salts, and vitamins C and E) [4,5]. In addition, studies suggest that bergenin has multifunctional properties, including antimalarial [6], antidiabetic [7], antioxidant [8], antiviral [9], and anti-inflammatory [10] activities, among others. The aim of this review is to describe the sources for the isolation of bergenin or its derivatives and its in vitro and in vivo biological and pharmacological activities.

**Table 1 biomolecules-13-00403-t001:** Plant species with bioavailability of bergenin.

Family	Scientific Name of the Plant	Part of Plant Used	Vernacular Plant Name	Native Location of the Plant	Reference
Acanthaceae	*Gendarussa vulgaris* Nees.	Aerial parts	Justicia gendarussa	India, Malaysia	[11]
Asclepiadaceae	*Streptocaulon griffithii* Hook. F.	Root	-	China	[12]
Araceae	*Arisaema franchetianum* Engl.	Tuber	-	Central Africa	[13]
Bombaceae	*Bombax malabaricum* L.	Flowers	Red silk cotton tree	India	[14]
Bignoniaceae	*Winter bignonia* (Ker Gawl.) Miers.	Rhizome	-	India	[15]
Caryophyllaceae	*Brachystemma calycinum* D.Don.	Aerial parts	-	Nepal	[16]
Connaraceae	*Connarus monocarpus*	Root	-	Peninsula of India	[17]
Convolvulaceae	*Rivea hypocrateriformis* (Desr.) *Choisy*	Stem	-	South Asia	[18]
Clusiaceae	*Garcinia malaccensis* Hook. F.	Stem bark	-	China	[11]
Compositae	*Pulicaria wightiana* C.B.Clarke	Aerial parts	Sonaphuli	India	[19]
Capprifoliaceae	*Brachystemma calycinum* D.Don.	Aerial parts	-	Nepal	[16]
Crassulaceae	*Crassula ovata* cv. Obligua.	Bark	Jade plant	South Africa and Mozambique	[8,20]
	*Rhodiola kirilowii* Reg	Leaf	Rose bush	Himalayas	[21]
Dilleniaceae	*Doliocarpus dentatus* (Aubl.)	Leaves	Thirsty vine	Mexico and tropical America	[22]
Dipterocarpaceae	*Dipterocarpus grandflorus* Blanco	Stem	Keruing Bilinbing	Asia	[2,23]
	*Dryobalonops aroniatica* C.F. Gaertn.	Stem bark and heartwood	Borneo camphor	Sumatra, Borneo, Peninsular Malaysia	[2,24]
	*Hopea utilis* (Bedd.) Bole	Leaf	Black kongu	India	[2,25]
	*Hopea sangal* Korth.	Leaf	Mersiput	Singapore	[2,26]
	*Shorea leprosula* PROSEA.	Heartwood	Red meranti	Indonesia	[27]
	*Shorea robusta* Roth.	Leaf and root	Sal	India	[28]
	*Vatica pauciflora* (Korth.) Blume.	Stem bark	-	Malaysia, Sumatera, Thailand and Vietnam	[29,30]
	*Vatica albiramis* Van.	Stem	-	Indonesia	[31]
	*Vatica bantamensis* (Hassk.) Benth. & Hook.F.	Leaf	-	Indonesia	[32]
	*Vatica diospyroides* S.	Stem	-	Malaysia	[33]
	*Vatica mangachpoi*	Leaf	Resak	Malaysia	[34]
	*Vateria indica* C.F. Gaertn.	Leaf, seed, and stem bark	-	Asia	[35]
Ebenaceae	*Diospyros sanza-minika* PROTA	Wood	Liberia ebony	Africa	[36]
	*Mussaenda erythrophylla* Schumach. & Thonn	Aerial parts	Pink mousseenda	Asia	[37]
Ericaceae	*Arctostaphylos uva-ursi* L.	Leafy shoot	Bearberry	Western America	[38]
Euphorbiaceae	*Fluggea virosa* (willd.) voigt	Aerial parts	Chinese waterberry	Saudi Arabia	[39]
	*Fluggea leucopyrus* Willd	Leaf	-	Asia	[40]
	*Fluggea microcarpa* Blume	Leaf	-	Southern Africa	[41]
	*Fluggea luvangetina*	Root	-	Paleotropics	[41]
	*Fluggea religiosa* L.	Bark	-	Paleotropics	[42]
	*Fluggea virens* L.	Bark	-	Paleotropics	[42]
	*Fluggea glomerata* L.	Bark	-	Paleotropics	[42]
	*Fluggea benghalensis* L.	Bark	-	Paleotropics	[42]
	*Glochidion obovatum* Siebold & Zucc.	Leaf	-	Japan	[43]
	*Glochidion obliquum* Siebold & Zucc.	Leaf	-	Japan	[44]
	*Excoecaria agallocha* L.	Leaf	Buta-Buta	Singapore	[45]
	*Mallotus japonicus* Müll. Arg.	Bark and cortex	Akamegashiwa	East Asia	[46]
	*Mallotus repandus* (Rottler) Müll. Arg.	Stem	Liana creeper	Tropical and Subtropical Asia	[47]
	*Mallotus anisopodus* Gagnep.	Aerial parts	-	Vietnam	[48]
	*Mallotus philippinensis* (Lam.) Mull.Arg.	Leaf and stem bark	Kamala tree	Asia and Australia	[49]
	*Mallotus roxburghianus* (Lam.) Müll.Arg.	Leaf	-	India	[50]
	*Mollotus oppositifolius*	Leaf	-	-	[51]
	*Macaranga peltata* (Roxb.) Müll.Arg	Bark	Chandada	India	[2,52]
	*Phyllanthus columnaris* Müll.Arg.	Root bark	-	Andaman Islands	[53]
	*Phyllanthus flexuosus* Müll.Arg.	Stem bark	-	China	[54]
	*Phyllanthus wightianus* Müll.Arg.	Whole plant	-	India	[54]
	*Securinega virosa* (Roxb.)	Leaf	Itachen-gado	Nigeria	[55]
	*Securinega melanthesoides* (F. Muell.) Airy Shaw	Leaf	-	Madagascar and Mascarene Islands	[56,57]
Fabaceae	*Peltophorum africanum* Sond.	Root	Weeping wattle	Africa	[6]
	*Peltophorum inerme* (Roxb.) Náves.	Flower	Yellow cassia	Southeast Asia	[58,59]
	*Peltophorum pterocarpum*	Flower	Yellow-flamboyant	Tropical Southeastem Asia	[60,61]
	*Peltophorum ferruginium* (DC.) Backer	Bark	Yellow Poinciana	Southeast Asia	[2]
	*Ciser microphyllum*	Aerial parts	-	Himalayas	[62]
	*Teramnus labialis* (L.f.) Spreng.	Aerial parts	Blue wiss	Tropical Africa	[63]
Gentianaceae	*Tripterospermum chinense* (Migo) Harry Sm.	Aerial parts	-	China	[2]
Hamamelidaceae	*Corylopsis coreana* Uyeki	Leaf	-	Eastern Asia	[64]
	*Corylopsis spicata*	Bark	Winter Hazel	Japan	[65]
	*Corylopsis willmottiae*	Whole plant	Chinese winter Hazel	China	[2]
Asteraceae	*Tridax procumbens* L.	Aerial	Bullweed	Tropical Americas	[66]
Humiriaceae	*Endopleura uchi* (Huber.) Cuatrec.	Bark	Uchi	Brazilian Amazon	[67]
	*Humiria balsamifera* (Aubl.) A.St.—Hil.	Aerial parts	Umiri de cheiro	Brazilian Amazon	[39,68]
	*Sacoglottis gabonensis* Urb.	Bark	Bitterbark tree	Tropical Africa and South America	[69,70]
Lythraceae	*Woodfordia fruticosa* (L.) Kurz	Stem	Fire-flame bush	Asia	[71]
	*Lagerstroemia speciosa* L.	Flowers	-	-	[14]
Leguminosae	*Caesalpinia decapetala* (Roth)	Root	-	China	[2]
	*Caesalpinia mimosoides* Lam.	Root	-	China	[2]
	*Caesalpinia pluviosa D*	Stem Bark	-	-	[2]
	*Caesalpinia digyna* Rottl.	Root	-	Eastern Himalayas, Assam, and West Bengal	[72]
	*Cenostigma macrophyllum* Tul.	Stem bark	Caneleiro	Brazil	[73]
	*Cenostigma gardnerianum* Tul.	Stem bark	Cinnamon	Brazil	[73]
	*Pentaclethra macrophylla* Benth.	Root	African bean	West and Central Africa	[74,75]
Malvaceae	*Brachystenuna calycinum* D. Don-GBIF	Root	-	China	[2]
	*Thespesia popunea* L.	Bark	-	-	[42]
Moraceae	*Ficus racemosa* L.	Bark	Red river fig	Australia and tropical Asia	[76]
Myrsinaceae	*Ardisia crenata* Sims.	Root	Coral bush; Blueberry.	East Asia	[77]
	*Ardisia colorata* Blume.	Fruit	Marlberry	China	[78]
	*Ardisia japonica* Blume.	Aerial parts	Passion fruit	East China, Japan, and Korea	[79]
	*Ardisia elliptica* Andr.	Root	Duck’s-eyes	West coast of India	[2]
	*Ardisia punctata* (Reinw.)	Root	Common Labisia	Southeast Asia	[2]
	*Ardisia pusilla* A. DC.	Root	Marlberry	Asia	[80]
	*Ardisia escalloniodes* S. and D.	Seed	Marlberry	Asia	[77]
	*Ardisia compressa* (Kunth.)	Seed	Marlberry	Asia	[77]
	*Ardisia mamillata* (Hance.)	Seed	Marlberry	Asia	[77]
	*Ardisia gigantifolia* Stapf.	Root	Marlberry	Asia	[81]
Oleaceae	*Olea dioca* Roxb.	Flowers	Marlberry	Asia	[14]
Pinaceae	*Pinus roxburghii* Charg.	Leaf	Marlberry	Asia	[82]
Ranunculaceae	*Cimicifuga foetida* L.	Rhizome	-	Europa and Siberia	[2]
	*Pulsatilla koreana* Mill.	Root	-	Korea	[2]
Rubiaceae	*Wendlandia thyrsoidea* (Roth) Steud.	Flowers	-	India	[2]
Saxifragaceae	*Astilbe chinensis* (Maxim.) Engl.	Rhizome	-	Japan	[83,84]
	*Astilbe rivularis* Buch. Ham.	Rhizome	River Astilbe	East Asian	[85]
	*Astilbe myriantha* Diels.	Rhizome	Rabbit ear	East Asian	[2,86]
	*Astilbe thunbergii* Miq.	Rhizome	-	Japan	[86]
	*Bergenia scopulosa*	Rhizome	-	China	[2,87]
	*Bergenia ligulata* Wall.	Leaf	-	Central Asia	[87]
	*Bergenia purpurascens* (Hook.f. e Thomson) Engl.	Rhizome	-	Asia	[82]
	*Bergenia stracheyi* (Hook. f. e Thomson) Engl.	Whole plant	-	Central Asia	[88]
	*Bergenia cordifolia* (Haw.) Sternb.	Rhizome	Siberian tea	Central Asia	
	*Boykinia lycoctonifolia* (Maxim) Engl.	Rhizome	-	North America and Asia	[83,84]
	*Peltiphyllum peltatum* L.	Rhizome	Indian-apple	Mato Grosso (Brasil)	[2]
	*Peltoboykinia watanabei* L.	Rhizome	-	Japan	[2]
	*Rodgersia sambucifolia*	Root	Elderberry	China	[2]
	*Rodgersia pinnata*	Rhizome	Bronze Peacock	China	[2]
	*Rodgersia aesculifolia* Bet.	Rhizome	-	Northern China	[1]
	*Saxifraga crassifolia* L.	Leaf	Siberian tea	Central Asia	[1]
	*Saxifraga melanocentra* FRANCH.	Leaf	-	China	[88]
	*Saxifraga stolonifera* Curtis.	Leaf	-	China	[89]
Sapindaceae	*Allophylus edulis var. edulis*	Leaf	Pigeon fruit	Latin America	[90]
Vitaceae	*Cissus javana* DC.	Root	Rex begonia vine	Thailand	[91]

## 2. Materials and Methods

### Data Collection

The bibliographic research methodology was adopted and conducted using the databases available for institutional access, i.e., Google Scholar, Medline, Pubmed andSciFinder. Regarding the latter, the research was carried out both using keywords and by searching for similarities with the chemical structure of the molecule bergenin. The following terms were used as descriptors of the bibliographic research: “bergenin”, “plant extracts”, “in vitro assays”, “in vivo assays”, “natural sources”, “chemical aspects”, “derivatives”, “antimalarial activity”, “antileishmanial activity”, “trypanocidal activity”, “antiviral activity”, “antibacterial activity”, “antifungal activity”, “anti-inflammatory activity”, “antinociceptive activity”, “antiarthritic activity”, “antiulcerogenic activity”, “antidiabetic/antiobesity”, “antiarrhythmic activity”, “anticancer activity”, “antioxidant activity”, “hepatoprotective activity”, “neuroprotective activity”, “cardioprotective activity”, and using the Boolean operators “AND” and “OR”. A total of 6550 papers, 6 master’s dissertations, and 2 doctoral theses were found, of which 210 papers, 2 master’s dissertations, and 2 doctoral theses published in the period from 1880 to 2022 were selected. The inclusion criteria were articles written in Portuguese, Spanish, or English, and their abstracts were available. Doctoral theses and master’s dissertations were also included. Papers, dissertations, and theses that studied biological and pharmacological activities of bergenin, natural derivatives of bergenin, semisynthetic derivatives of bergenin, and plant extracts with phytochemical confirmation that bergenin is one of the constituents were selected. The exclusion criteria were studies that did not refer to at least one of the research themes and papers that had been retracted.

## 3. Natural Sources of Bergenin

The bergenin molecule has been isolated/identified in at least 112 plant species distributed in 34 plant families: Euphorbiaceae (23 species belonging to 6 genera), Saxifragaceae (18 species belonging to 7 genera), Dipterocarpaceae (12 species belonging to 6 genera), Leguminosae (7 species belonging to 5 genera) Humiriaceae (3 species belonging to 3 genera), Fabaceae (6 species belonging to 3 genera), Myrsinaceae (10 species belonging to 1 genus), Hamamelidaceae (3 species belonging to 1 genus), Crassulaceae (2 species belonging to 2 genera), Lythraceae (2 species belonging to 2 genera) Connaraceae (1 species), Convolvulaceae (1 species), Malvaceae (2 species belonging to 2 genera), Ranunculaceae (2 species belonging to 2 genera), Ebenaceae (2 species belonging to 2 genera), Clusiaceae (1 species), Acanthaceae (1 species), Ericaceae (1 species), Araceae (1 species), Caryophyllaceae (1 species), Compositae (1 species), Asclepiadaceae (1 species), Asteraceae (1 species), Capprifoliaceae (1 species), *G*entianaceae (1 species), Bombaceae (1 species), Dilleniaceae (1 species), Oleaceae (1 species), Pinaceae (1 species), Rubiaceae (1 species), Sapindaceae (1 species), Vitaceae (1 species) and Moraceae (1 species) (Table 1). In terms of geographical distribution, most of the species with bioavailability of bergenin are located on the Asian continent, with emphasis on China. The concentration of bergenin in tissues or organs in the plant will depend on each species. However, four most frequently stand out: rhizomes, roots, leaves, and trunk bark (Table 1).

## 4. Chemical Aspects of Bergenin

Bergenin (IUPAC name: 4-methoxy-2-[(1*S*,2*R*,3*S*,4*S*,5*R*)-3,4,5,6-tetrahydro-3,4,5-trihydroxy-6-(hydroxymethyl)-2*H*-pyran-2-yl]-*α*-resorcylic acid) (Figure 1) is a *C*-glycosylated derivative of 4-*O*-methyl gallic acid [92,93]. It is a hydrolyzable phenolic glycoside that is obtained as a colorless crystal, has low solubility in water, degrades easily in basic solution (for example, Dimethyl sulfoxide in the concentration of 0.01–5%), and its stability depends mainly on storage conditions (best storage condition at −80 °C) [94]. It was discovered in 1880 by Garreau and Machelart [92]. The first proposals on the structure of bergenin were provided by Tschitschibabin et al. (1929). First, a chemical structure with only two rings and one aliphatic chain was proposed. Later, in 1950, the structure was revised by Shimokoriyama and proposed with three rings of six members [92,93] (Figure 1).

The biosynthesis of bergenin is directly related to the biosynthesis of gallic acid, which originates in the combination of erythrose-4-phosphate with phosphoenolpyruvate, which leads to the formation of 3-dehydroquinic acid. Dehydration of the latter leads to the formation of 3-dehydroshikimic acid, which in turn is converted to gallic acid through oxidation and enolization reactions. Other alternative routes would be the degradation of the side chain of hydroxycinnamic acids or through the condensation of an acetyl-CoA group with three malonyl-CoA units. Regarding the *C-*glycosylation step, previous experiments have shown contrasting results since the use of isotopically labeled [7-^14^C] benzoic acid and hydroxycinnamic acids indicated that *C-*glycosylation would occur in C_6_-C_3_ compounds, to the detriment of the C_6_-C_1_ derivative such as benzoic acid [95]. On the other hand, the use of the molecule labeled D-[U-^14^C] glucose in the presence of labeled gallic acid indicated that the *C-*glycosylation step occurs preferentially in the C_6_-C_1_ derivative [96]. Subsequently, the condensation of the glycosidic portion with the carboxylic acid of the gallic acid portion occurs, which leads to the formation of the lactone portion observed in the bergenin structure. Finally, more recent studies have suggested that the *C-*glycosylation step occurs prior to the *O-*methylation of the phenolic hydroxyl of the gallic acid-derived portion by *O-*methyltransferases [97].

Several bergenin derivatives have been isolated from plants, but not all showed marked biological activities. The main substances derived from bergenin with this type of potential are demethylated analogs, or those which are esterified with phenolic acids (e.g., gallic acid). Among the natural derivatives of bergenin, the following stand out: riverbergenin A isolated from the trunk of *R. hypocrateriformis* [18], norbergenin from the leaves of *A. japonica* [79], 11-*O*-galloylbergenin from the rhizome of *A. gigantifolia* [81] 11-*O*-veratroylbergenin from the rhizome of *A. gigantifolia* [81], 4-*O*-galloylnorbergenin isolated from the stem bark of *M. japonicus* [98], 8-*O*-methylnorbegenin from all parts of *S. stolonifera* [99], 11-*O*-acetylbergenin from the aerial parts of *F. virosa* [100], 11-*O*-vanilloylbergenin from the roots of *A. crenata* [101], 11-*O*-*p*-hydroxybenzolynorbergenin, 4-*O*-(3′-*O*-methylgalloyl)norbergenin and 4-*O*-syringoylnorbergenin from the bark of the stem of *D. sanza-minika* [36,102]. In addition to these, some semisynthetic derivatives have been designed in order to increase their pharmacological potential, for which several modifications have been proposed. Figure 2 lists all the natural and semisynthetic derivatives of bergenin cited in this review.

## 5. Biological and Pharmacological Activities of Bergenin

### 5.1. Antimalarial Activity

Bergenin, or its derivatives and extracts containing bergenin, have been reported to have antimalarial activity in vitro and in vivo studies. To date, about 15 in vitro and 5 in vivo studies on this topic have been published (Table 2).

Singh et al. [39] reported the in vitro antimalarial activity of the crude extract and fractions derived from *F. virosa* against *Plasmodium falciparum* using the nucleic acid dye SYBR Green I-based fluorescence assay (MSF). In the study, different concentrations of the extract (0.1 to 100 µg/mL) were incubated with chloroquine-sensitive (3D7) and chloroquine-resistant (K1) strains of *P. falciparum,* and subsequently, the plates were examined at 485.20 nm excitation and 530.20 nm emission for fluorescence units (RFUs) per well using a fluorescence reader (FLX800, BIOTEK). The findings indicated that the extracts/fractions significantly inhibited the sensitive (3D7) and resistant (K1) strains of *P. falciparum*. The crude extract presented an IC_50_ of 2.35 µg/mL for 3D7 and 4.73 µg/mL for K1, while the fractions presented an IC_50_ of 1.73 to 8.61 µg/mL for 3D7 and 2.32 to 20 µg/mL for K1 against 5.5 and 2.54 nM of the positive control (chloroquine), respectively [39]. Other studies with the same approach, using extracts with bergenin as one of the constituents, had satisfactory results in vitro: ethanolic and methanolic extract of stems and leaves of *H. balsamifera* against a culture of BHz 26/28 of *P. falciparum* resistant to chloroquine, IC_50_ ranging from 8.37 to 49.65 µg/mL [13]; ethanolic extract of the rhizome of *W. bignonia* against strains of *P. falciparum* RKL-9 2, IC_50_ of 5 µg/mL [15]; ethanolic extract of *B. ciliata* rhizome against *P. falciparum* strains RKL-9 and MRC-2, IC_50_ of 5 µg/mL [15]; methanolic extract of *D. Sanza-Minika* stem bark against *P. falciparum* K1, IC_50_ of 0.6 µg/mL [36].

The IC_50_ of pure bergenin isolated from different parts of plant species (Table 2) was determined against *P. falciparum* in vitro, with the following results: 2.41–14.1 µg/mL [37,38,40,50,58,103]. The mechanism of action by which bergenin inhibits the growth of the parasite in vitro is not yet well known, but it is suggested that it is triggered by the inhibition of polymerization of the heme group of the parasite in a similar way to the mechanism of action of artemisinin since both are lactones of the same family [39,104]. In other words, in the first step, bergenin is activated by heme or a free iron (II) ion and produces free radicals and cytotoxic species. In the second step, these species react with the specific protein associated with the parasite’s membrane, thus causing its death [104].

11-*O-***G**alloylbergenin, a bergenin derivative isolated from the root of *B. ligulata*, was tested against *P. falciparum* and showed significant activity against a resistant *P. falciparum* strain (CQS D10) and exhibited good activity at low concentrations, with an IC_50_ value of 2.34 µg/mL against a value of 28.07 nM for CQ (chloroquine) as a positive control [105]. These results are similar to the findings of a study in which 11-*O*-galloylbergenin was isolated from all parts of the plants of the species *M. philippensis* [49]. Other derivatives of natural bergenin had their IC_50_ determined against *P. falciparum*, namely, 4-*O-*(3′-*O-*methylgalloyl) norbergenin (IC_50_ of 0.6 µg/mL) [36], 4-*O*-galloylnorbergenin (IC_50_ 3.9 µg/mL) [36], and 11-*O*-*p*-hydroxybenzoylnorbergenin (IC_50_ of 4.9 μg/mL) [36]. 

The in vivo antimalarial activity of bergenin was also evaluated (Table 2). Experimental models have used animals such as rats and mice infected with *P. berghei* [106]. *P. berghei* is a parasite that infects rodents and was discovered in the 90s in Congo. Since then, it has been widely used for experimental infections, which are considered models for biological and therapeutic studies [106]. Thirteen mice experimentally infected with *P. berghei*, with parasitemia of 35%, were treated with 800 mg/kg/day (100 µL twice daily) of bergenin (isolated from the leaves of *R. aesculifolia*) for 4 to 6 days intragastrically, and the findings showed that bergenin decreased parasitemia from 35 to 27%. These data indicate that bergenin suppresses the growth of *P. berghei* in vivo [6]. Singh et al. [39] experimentally induced *P. berghei* infection intraperitoneally in mice and, after 7 days of infection, treated with bergenin (isolated from *F. virosa* leaves) at concentrations of 25 to 100 mg/kg for 8 days, it caused an 85.13% suppression of parasitemia on the eighth day at a concentration of 100 mg/kg bergenin. These results are similar to the findings of Gorky et al. [15]. 

On the other hand, Da Silva et al. [107], in vitro experiments using bergenin (isolated from *H. balsamifera* leaves) against *P. falciparum,* found no activity (Table 2), but in vivo experiments found moderate efficacy in inhibiting the growth of *P. berghei* (IC_50_ of 146.87 mg/kg), which suggests that more studies should be done to determine the concentrations for successful parasitic inhibitions.

**Table 2 biomolecules-13-00403-t002:** Summary of studies regarding the antimalarial activity of bergenin and its derivatives. In vitro tests were carried out against *P. falciparum* and in vivo (in mice) against *P. berghei*.

Compound Name	Source of Isolation	Assay Type, IC_50_	Cytotoxicity		Reference
			CL	CV	
Bergenin	Ethanolic extract of the rhizome of *W. bignonia*	in vitro, 5.0 µg/mL	HeLa and dermal fibroblasts	NC	[15]
Bergenin	Leaves of *F. virosa*	in vitro, 8.07 µg/mL	Murine intraperitoneal macrophages	NC	[39]
Bergenin	Methanolic extract of the stem bark of *D. sanza-minika*	in vitro, 0.6 µg/mL	ND	ND	[36]
Bergenin	Leaves of *H. balsamifera*	in vitro, SA	ND	ND	[107]
Bergenin	Ethanolic extract of the rhizome of *B. ciliata*	in vitro, 5.0 µg/mL	HeLa and dermal fibroblasts	NC	[15]
Bergenin	Rhizome of *B. ciliata*	in vivo -mice, 50 mg/kg	HeLa and dermal fibroblasts	NC	[15]
Bergenin	Leaves of *R. aesculifolia*	in vitro, 14.1 µg/mL	HeLa and HepG2	NC	[6]
Bergenin	Leaves of *R. aesculifolia*	in vivo -mice, 800 mg/kg)	HeLa and HepG2	NC	[6]
Bergenin	Roots of *B. ligulata*	in vitro, 2.41 µg/mL	ND	ND	[105]
Bergenin	Whole plant of *M. philippensis*	in vitro, 6.92 µM	ND	ND	[49]
11-*O*-Galloylbergenin	Roots of *B. ligulata*	in vitro, 2.34 µg/mL)	ND	ND	[108]
4-*O*-(3′-Methylgalloyl)norbergenin	Methanolic extract of the stem bark of *D.sanza-minika*	in vitro, 0.6 µg/mL	ND	ND	[36]
4-*O*-Galloylnorbergenin	Methanolic extract of the stem bark of *Diospyros sanza-minika*	in vitro, 3.9 µg/mL	ND	ND	[36]
11-*O*-*p*- Hydroxybenzoylnorbergenin	Methanolic extract of the stem bark of *Diospyros sanza-minika*	in vitro, 4.9 µg/mL	ND	ND	[36]
11-*O*-Galloylbergenin	Whole plant of *M. philippensis*	in vitro, 7.85 µM	ND	ND	[49]
Bergenin	Extract of the aerial part of *M. erythrophylla*	in vitro, 7.43 µg/mL	Raw 264.7 macrophage cells	NC	[37]
Bergenin	Leaves of *F. virosa*	in vivo -mice,100 mg/Kg)	Murine intraperitoneal macrophages	NC	[49]
Bergenin	Leaves of *H. balsamifera*	in vivo -mice, 146.87 mg/kg	ND	ND	[107]
Bergenin	Ethanolic extract of the rhizome of *W. bignonia*	in vivo -mice, 50 mg/kg	HeLa and dermal fibroblasts	NC	[15]

Abbreviation: NC: no cytotoxicity, ND: no cytotoxicity determined, NA: no activity, CL: cell line, CV: cell viability.

### 5.2. Antileishmanial Activity

The antileishmanial activity of bergenin has been reported in six in vitro studies and in one in vivo study. Crude extracts/fractions of the aerial parts of *M. erythrophylla* have shown an inhibitory effect against *Leishmania donovani*, which is the etiological agent of visceral leishmaniasis, in a resazurin colorimetric test [37]. The extracts were incubated together with visceral leishmaniasis 1S (MHOM/SD/62/1S) promastigotes at concentrations of 0.16 to 100 µg/mL, in triplicate, followed by the addition of resazurin and amphotericin B (10–0.016 µg/mL), was used as the positive control. The results indicated that the crude extract had a moderate effect on antileishmanial activity (IC_50_ of 61.6 µg/mL), and the hexane fraction showed good antileishmanial activity (IC_50_ of 31.06 µg/mL) when compared with the reference drug, which was the positive control amphotericin B (IC_50_ of 0.11 µM) [37]. Kaur and Kaur [103] reported antileishmanial activity of the ethanolic extract of *B. ligulata* root against *L. donovani* after finding satisfactory parasitic inhibition (IC_50_ of 22.70 µg/mL), and phytochemical data revealed that the main compound in *B. ligulata* is bergenin, which suggests that the antileishmanial activity of these extracts is due to this compound [103]. Keshav et al. [62] reported that hydroxyethanolic extract of *C. microphyllum* showed good activity against sensitive and resistant strains of *L. donovani*, with an IC_50_ of 14.40 µg/mL and 23.03 µg/mL, respectively. Kabran et al. [51] reported that bergenin isolated from leaves of *M. oppositifolius* had an effect against *L. donovani* (IC_50_ of 73.3 µM).

Bergenin isolated from aerial parts of *M. erythrophylla* showed an inhibitory effect against *L. donovani* in resazurin colorimetric tests [37]. Bergenin was incubated together with protozoa, in triplicate, at concentrations of 0.08 to 50 µg/mL. Amphotericin B (10–0.016 µg/mL) was used as a positive control; the results showed the promising effects of bergenin against *L. donovani* (IC_50_ of 53.7 µM) [37].

Antileishmanial activity was also evaluated in vivo. Kaur and Kaur [103] reported that ethanolic extract of *B. ligulata* root had an antileishmanial effect in inbred BALB/c mice infected intracardially with 10^7^ promastigotes. In the study, 48 mice were divided into two groups, an uninfected group, and an infected group, and the infected groups had inflammation in the liver and spleen triggered by the recruitment of the inflammatory cytokines: interleukin 12 (IL-12), interleukin 4 (IL-4), interleukin 10 (IL-10) and interferon-gamma (IFN- γ). The infected group was treated with the ethanolic extract of *B. ligulata* orally at concentrations of 500 and 1000 mg/kg for 15 days, and the mice were euthanized at 1, 7, 14, and 21 days post-treatment. The spleen and liver were removed, then centrifuged, and the homogenate was used to quantify the parasitemia and determine the immune response by the parasite-specific enzyme-linked IgG1 and IgG2a isotopes, delayed-type hypersensitivity (DTH) responses, and the effect of recruited cytokines. The results indicated that the treatment with the extracts rich in bergenin at both concentrations tested significantly reduced the parasitic load, corresponding to 91.1% and 95.6%, respectively. The treatment with the extracts induced the IgG2a antibody response, which is considered an indicator of the Th1 type of immune response, thus contributing to parasitic reduction. The treatment also triggered inhibition of the Th2 activation pathway by reducing the production of inflammatory cytokines, especially IFN- γ at 5001.61 to 175.21 pg/mL, which contributed to the reduction of inflammation of the internal organs of infected mice [103].

### 5.3. Trypanocidal Activity

The trypanocidal activity was mainly evaluated using bergenin-rich extracts, with four in vitro studies and one in vivo study. Ethanolic extract of leaves of *C. pluviosa* rich in bergenin showed trypanocidal activity against *Trypanosoma cruzi* in vitro [72]. The extracts were incubated at concentrations of 10 to 500 µg/mL together with the parasitic suspension of *T. cruzi* in the infective trypomastigote form (2 × 10^5^/0.1 mL) in LIT (liver infusion tryptose) culture medium and, after 24 h, the suspension was microscopically quantified. The results indicated that the ethanolic extract of *C. pluviosa* showed good activity against the development of trypomastigotes of *T. cruzi* (IC_50_ of 55 µg/mL). Nyasse et al. [109] reported that bergenin isolated from *F. virosa* leaves also exhibited inhibitory activity in the growth of *T. brucei* in the bloodstream (trypomastigotes), with an IC_50_ value of 1 mM. Growth inhibition is achieved by inhibiting three glycolytic enzymes of the parasite: GAPDH (glyceraldehyde-3-phosphate dehydrogenase), PFK (phosphofructokinase), and PGK (phosphoglycerate kinase), which leads to the death of the parasite. Melos and Echearria [110] reported that trypanosomatids are highly dependent on glycolysis for ATP production, and, as many glycolytic enzymes have their own characteristics, they are considered to be potential targets for new chemotherapeutic agents.

Nyunt et al. [111] reported the trypanocidal activity of 11-*O-*acetylbergenin (isolated from the methanolic extract of *V. repens*) against *Trypanosoma evansi* trypomastigotes in vitro. 11-*O-*acetylbergenin was incubated in 96-well plates along with *T. evansi* in the trypomastigote form. Subsequently, the activity was determined by counting the number of parasites using a Neubauer hemocytometer. The findings indicated that 11-*O-*acetylbergenin inhibited (IC_50_ of 0.17 mM) the growth of *T. evansi*, thus suggesting trypanocidal activity.

The trypanocidal activity of the ethanolic extract of the stem bark of *S. gabonensis* was investigated in rats infected with *Trypanosoma congolense*. Twenty albino rats in four groups (A, B, C, and D) of five rats were injected intraperitoneally with 0.5 mL of blood infected with *T. congolese*. A further ten rats (E and F) were not infected with the parasite. All the infected animals developed the following clinical manifestations: varying degrees of lethargy, mucosa, rough fur, reduced appetite, and depression. Groups A, B, and C were treated with the ethanolic extract of *S. gabonensis* from the stock solution orally via water intake. Group A received 0.5 mg/kg, Group B received 1 mg/kg, and Group C received 3.5 mg/kg; however, Group D did not receive any treatment, Group E received treatment with the stock solution (0.5 mg/kg) and Group F did not receive any treatment. The results show that the treated infected groups (A, B, and C) had a mortality rate of 40 to 80% with 13 to 32 days of survival, while the infected Group (D), which received no treatment, had a mortality rate of 100% with 12 to 23 days of survival. The other negative control groups did not suffer any impact. These results suggest that treatment with ethanolic extract of the stem bark of *S. gabonensis*, which is rich in bergenin, reduced the growth of *T. congolense* and prolonged the lifespan of the infected rats [69].

### 5.4. Antiviral Activity

Bergenin has shown promise in studies focused on antiviral activity since it was discovered that extracts of species of the genus *Bergenia* exert immunostimulating activity in response to viruses and pathogenic microorganisms after their invasion of biological systems [112]. In this context, it is reported that bergenin can induce the transmembrane glycoprotein CD64, which has an important role in the humoral immune response, using an Fc receptor that binds to monomeric antibodies such as IgG [113]. To date, about 8 in vitro studies have been performed, though no study evaluated in vivo conditions.

The methanolic extract of *A. rivularis* showed an antiviral effect against herpes simplex virus type 1 (HSV-1) and influenza A via a dye-uptake assay in HSV-1/Vero cell systems and influenza A virus/MDCK [85]. The cells were incubated together with the viruses in different concentrations of extracts (6.25 to 100 µg/mL), and the results revealed that the extracts exhibited good antiviral activity with consequent destruction of HSV-1 and influenza A viruses IC_50_ of 6.25 µg/mL. These results suggest that extracts of *A. rivularis* exerted antiviral activity due to the presence of secondary metabolites since phytochemical investigations of the plant showed the presence of bergenin, flavonoids, and terpenoids [114]. Under the same conditions, Rajbhandar et al. [9] evaluated bergenin isolated from the rhizome of *A. rivularis*; however, they only tested the antiherpes activity, and the results were similar. Bergenin exhibited good activity against the herpes virus (IC_50_ of 6.25 µg/mL), proving that bergenin has this potential and can be explored further. 

Zuo et al. [88] evaluated bergenin isolated from the aqueous ethanolic extract of the aerial parts of *S. melanocentra* against the NS3 serine protease of the hepatitis C virus (HCV) in an enzyme-linked immunosorbent assay (ELISA). HCV is an enveloped virus with positive-stranded genomic RNA that encodes a polyprotein that is cleaved by the NS3 serine protease, thus allowing the development of HCV [115] and, therefore, has been targeted for the development of new therapies for HCV infection. In the study, bergenin was incubated along with NS3 serine protease, and the results indicated that bergenin significantly inhibited NS3 serine protease (IC_50_ of 27.7 μg/mL).

Piacente et al. [116] evaluated the methanolic extract of aerial parts of *A. japonica* (the main constituent of bergenin) against the HIV virus in vitro and showed a moderate effect. However, when they tested bergenin and norbergenin isolated from the same extracts, both compounds showed weak antiHIV activity. The methanolic extract of the roots and bark of *P. africanum* was evaluated for its activity against HIV-1 reverse transcriptase (RT) [117]. Drugs that fight the HIV virus act as inhibitors of the reverse transcriptase, integrase, and protease enzymes. However, by inhibiting these enzymes, drugs prevent the multiplication of the virus in host cells. Reverse transcriptase is an enzyme that performs reverse transcription by producing DNA from RNA. It is also called RNA-dependent DNA polymerase. Therefore, the extracts incubated with this enzyme showed an inhibitory effect (IC_50_ of 3.5 µg/mL) against the DNA-dependent RNA polymerase activity (RDDP) of RT; however, bergenin isolated from the same extract had no effect against the same enzyme at concentrations tested up to 100 µM [117].

### 5.5. Antibacterial Activity

The antibacterial activity of bergenin has been reported in 15 in vitro studies and 4 in vivo studies. Liu et al. [82] reported that the methanolic extract of leaves of *B. purpurascens* showed antibacterial activity against strains that commonly cause respiratory infections: *Streptococcus pneumonia**e***, *Haemophilus influenzae*, *Klebsiella pneumoniae*, *Escherichia coli*, *Staphylococcus aureus,* and *Enterobacter cloacae*. The bacteria were incubated in triplicate along with the extracts at concentrations of 0.8 to 512 µg/mL using the agar well diffusion method, with resazurin as the dye added to each well. The results showed that the extract of *B. purpurascens* inhibited the growth of all strains, with IC_50_ ranging from 27 to 280 µg/mL. Other studies with the same approach using extracts that have bergenin had satisfactory results in vitro: methanol extracts of flowers of four angiosperm plant species *Wendlandia thyrsoidea*, *Olea dioica*, *Lagerstroemia speciosa,* and *Bombax malabaricum* against *S. aureus*, *Bacillus cereus*, *Vibrio cholerae* and *E. coli*, IC_50_ ranging from 1.2 to 3.8 µg/mL [14]; ethanol extract of *C. corean* against resistant strains of *S. aureus*, IC_50_ of 20 µg/mL [118]; methanolic extract of the rhizome of *B. ligulata* against *E. coli* and *S. aureus*, IC_50_ of 250 µg/mg [119]; methanolic extract of the rhizome of *A. rivularis* against *E. coli*, IC_50_ of 100 µg/mL [120]. 

Bergenin isolated from the ethanolic extract of *P. roxburghii* was tested in vitro using the agar well diffusion method [82] against Gram-positive (+) (*S. aureus and E. faecalis*) and Gram-negative (-) bacteria (*P. aeruginosa*, *K. pneumoniae*, *S. typhi*, *E. coli*, *Acenatobacter* sp., and *Proteus* sp.) [121]. The findings indicated that bergenin had a significant effect against the bacteria *E. coli* (−), *K. pneumoniae* (−), and *P. aeruginosa* (−) (IC_50_ ranging from 0.78 to 1.56 µg/mL); however, it had a weak effect against the bacteria *S. typhi* (−), *Acenatobacter* sp. (−), *E. Proteus* sp. (−), *E. phaeacalis* (+) and *S. aureus* (+) (IC_50_ ranging from 3.125 to 6.25 µg/mL) [122]. Nyemb et al. [123] reported that bergenin isolated from *C. populnea* roots had an inhibitory effect against four strains of Gram-negative bacteria in vitro: *S. typhi* (ATCC6539), *S. typhi* (isolated), *P. aeruginosa* (ATCC9721) and *E. coli* (isolated), IC_50_ ranging from 8 to 64 µg/mL. These results suggest that bergenin is able to easily cross the complex and multilayer lipopolysaccharide cell walls of Gram-negative bacteria. In doing so, it weakens them, causing lysis and consequent death [123]. However, some studies with bergenin had negative results [124,125]. Silva et al. [67] reported that bergenin isolated from the ethyl acetate fraction of *E. uchi* bark had no inhibitory effect against Gram-positive and Gram-negative bacteria. Similarly, Raj et al. [125] reported that bergenin isolated from crude methanolic extract of *P. pterocarpum* flowers had no inhibitory effect against bacteria.

Semisynthetic derivatives of bergenin isolated from the root of *P. dubium* [122] showed antibacterial activity in vitro using the agar diffusion method. 8,10-dihexyl-bergenin (1a), 8,10-didecyl-bergenin (1b), 8,10-ditetradecyl-bergenin (1c), 8,10-dimethylbergenin (1d), 8-methylbergenin, and 8,10-dioctyl-bergenin (1e) (Figure 2) exhibited antibacterial activity against *S. aureus*, *B. subtilis*, *E. coli*, and *K. pneumoniae*, with a minimum inhibitory concentration (MIC) of 5.1–6.2 mM [126], which suggests that pure bergenin can be transformed into a potent antibacterial agent [126]. 8,10-dibenzoylbergenin (1f) (IC_50_ of 125 µg/mL) was shown to be better antibacterial than bergenin (IC_50_ of 250 µg/mL) in an in vitro assay against *S. aureus*, which indicates that it is also a potential antibacterial derivative of bergenin [127].

Antibacterial activity was also determined in vivo. Liu et al. [82] evaluated the antibacterial activity of the methanolic extract of *P. roxburghii* (bergenin being the main constituent) against *S. aureus* in a neonatal Wister rat model. *S. aureus* is a bacterium of the Gram-positive cocci group that is part of the human microbiota, but that can cause diseases ranging from a simple infection, such as pimples and boils, to more serious ones, such as pneumonia and meningitis. Ten newborn rats were divided into four groups in which they received the treatments together with the *S. aureus* bacteria orally for four days in a row: Group I received saline, Group II received normal saline plus 2 mL of *S. aureus*, Group III received 50 mg of the extract plus 2 mL of *S. aureus* and Group IV received 100 mg of the extract plus 2 mL of *S. aureus*. The survival and weight of the rats were monitored for 9 days. The results showed that the survival rate of Group I was 80%. Infection with *S. aureus* led to mortality and reduced the survival % to 34.28% in Group II. However, in Groups III and IV, the percentage of survival was 48.57 and 60%, respectively, suggesting that extracts of *B. purpurascens* could reduce mortality caused by infection with *S. aureus* in newborn rats [82]. Kumar et al. [128] evaluated bergenin as an adjuvant immunotherapeutic agent for tuberculosis in mice. Mice were infected with 15 mL of *M. tb* H37Rv suspension using a Madison aerosol chamber, which targets the lungs and spleen. Days later, they were treated intraperitoneally with 4 mg of bergenin for 45 days. Then, the mice were euthanized, and the spleen and lung organs were removed. The results indicated that adjuvant therapy with bergenin protects mice against tuberculosis (IC_50_ 4 mg). It is suggested that this treatment induces the adaptive immune response by inducing memory T cells (CD8+ T, CD62L1o, and CD44hi), which can provide lasting protection against pathogens. This knowledge could be used to potentiate the BCG vaccine for TB [128]. Dias et al. [129] reported that the ethyl acetate leaf extract (EALE) of *H. balsamifera* (bergenin being the main constituent) had an effect on the inhibition of *S. aureus* in mealworms. The larvae were divided into three groups: Group I larvae infected with a lethal dose of *S. aureus* without treatment, Group II uninfected larvae plus treatment with 3 mg of EALE, and Group III larvae infected with a lethal dose of *S. aureus* plus treatment with 3 mg of EALE. The findings indicated that Group I had a mean lifespan of 1 day, Group II had no decrease in its lifespan and Group III had a prolonged lifespan (5 to 6 days), and, at the end of the evaluation, 50% of the larvae were still alive, indicating that EALE) inhibits the growth of *S. aureus* [130].

### 5.6. Antifungal Activity

The antifungal activity of bergenin has been reported in five in vitro studies. Bergenin isolated from the ethyl acetate fraction of *E. uchi* bark was tested against *Candida albicans*, *Candida tropicalis*, *Candida guilliermondii*, *Aspergillus flavus*, *Aspergillus nidulans,* and *Aspergillus niger* using the well agar diffusion method [67]. The fungi were coated in triplicate along with bergenin in different concentrations, and resazurin was added to each well as a dye. The results indicated that the presence of bergenin inhibits the growth of the yeasts *C. albicans* (IC_50_ of 14.9 µM), *C. guilliermondii* (IC_50_ of 28.8 µM), and *C. tropicalis* (IC_50_ of 14.9 µM) but has lower activity against the filamentous fungi *A. falvus* (IC_50_ of 1093.0 µM), *A. niger* (IC_50_ of 476.1 µM), *A. nidulans* (IC_50_ of 951.9 µM) [67]. Raj et al. [125] reported that bergenin isolated from the crude methanolic extract of *P. pterocarpum* flowers had an inhibitory effect against *Trichophyton mentagrophytes* (IC_50_ of 250 µg/mL), *Epidermophyton floccosum* (IC_50_ of 500 µg/mL), *Trichophyton rubrum* (IC_50_ of 500 µg/mL), *Aspergillus niger* (IC_50_ of 500 µg/mL), and *Botrytis cinerea* (IC_50_ of 250 µg/mL). Rolta et al. [119] reported that bergenin isolated from the methanolic extract of the rhizome of *B. ligulata* had an inhibitory effect against two *Candida* sp. resistant strains: *C. albicans* (MTCC277) and *C. albicans* (ATCC90028), both with IC_50_ of 250 µg/mL. Pavithra et al. [14] reported that methanol extracts of flowers from four angiosperm plant species, *W. thyrsoidea*, *O. dioica*, *L. speciosa,* and *B. malabaricum* hadn’t inhibitory effect against *C. albicans*. These results suggest that further studies should be carried out in order to determine a good inhibitory concentration, and this observation was also made by Nyemb et al. [123].

### 5.7. Anti-inflammatory Activity

Several studies have evaluated the anti-inflammatory activity of bergenin, 16 in vitro and 12 in vivo. In all studies, bergenin showed that it has optimal cell viability (Table 3). Nunumura et al. [131] reported the anti-inflammatory activity of bergenin (0.01–1000 µM isolated from the trunk bark of *E. uchi*) against three enzymes in vitro, namely, cyclooxygenase-1 (COX-1), cyclooxygenase-2 (COX-2) and phospholipase A2 (PLA2). The effects of bergenin against COX-1 and COX-2 were determined by measuring the levels of prostaglandin E2 (PGE2), and the effects against PLA2 were determined by measuring its concentration by means of HPGP (1 hexadecanoyl-2-10-pyrenyldecanoyl-sn-glycero-3-phosphoglycerol). The results showed that bergenin selectively inhibited COX-2 (IC_50_ of 1.2 µM), though it was poorly active against PLA2 (IC_50_ of 156.6 µM) and was not able to inhibit COX-1 (IC_50_ of 107.2 µM). These results corroborate the findings of Li et al. [129] (IC_50_ of 100 µM) and De Oliveira et al. [132] (IC_50_ of 1.2 µM). However, Jachak et al. [66] had a satisfactory result in COX-1 inhibition (IC_50_ of 70.54 µM) and demonstrated that bergenin inhibits both COX-1 and COX-2. COX-1 and COX-2 enzymes convert arachidonic acid to prostaglandin E2; thus, its inhibition can cause relief in inflammatory symptoms, as occurs with ibuprofen and celecoxib [133]. Bergenin showed an anti-inflammatory effect in human immortalized keratinocytes (HaCaT) [134] induced by interferon (TNF-α). These were treated with bergenin at concentrations of 0.01 to 200 µM, and the results show that bergenin (IC_50_ of 50 µM) triggers activation of the Nrf2 (erythroid-derived nuclear factor 2) pathway and inactivation of the NF-κB (nuclear factor k-b) pathway, inducing upregulation of IL-6 and IL-8 in HaCa. In other words, it reduces the levels of IL-6 and IL-8 and demonstrates an anti-inflammatory effect since blocking the expression levels of these two cytotoxins reduces inflammation. These results are similar to the findings of Chen et al. [135] and suggest that bergenin (IC_50_ of 100 µM) inhibits the pro-inflammatory response induced by TNF-α by blocking the NF-κB signaling pathway.

Shah et al. [126] isolated bergenin from the crude extract of *M. philippenensis* and then subjected it to chemical derivatization, obtaining 16 different synthetic derivatives. They subsequently tested its anti-inflammatory activity in vitro. Bergenin and its derivatives were incubated with macrophages treated with LPs (lipopolysaccharides) to determine whether bergenin could activate the iNO pathway for nitric oxide (NO) production and inhibit TNF-**α** production to normalize the cells. The results showed that only compounds 1g (IC_50_ of 322.1 µM) and 1h (IC_50_ of 253.2 µM) showed significant activity for NO production (Figure 2). Jungo et al. [136] also obtained good results, which suggests that the suppression of inflammatory cytokines may be associated with excessive production of NO (IC_50_ of 30 µM).

In vivo studies have determined the anti-inflammatory activity of bergenin in edema of the paw, ear, intestine, lungs, and mammary glands of experimentally induced rats and mice. Male BALB/c mice were experimentally induced into acute pulmonary injury/edema by intranasal inhalation of LPs. The mice showed histological changes, with increases in the activity of myeloperoxidases (MPO) in lung tissues and inflammatory cells (BALF) and decreases in cytokines (in BALF and serum) and, after 12 h, were treated with bergenin at concentrations of 50, 100 and 200 mg/kg. The results indicated that bergenin (IC_50_ of 50 mg/kg) repaired the injured tissue and normalized the edema at all concentrations tested: decreased MPO activity (decumulated neutrophils in lung tissues), decreased inflammatory cells (neutrophils and macrophages in BALF), increased production of inflammatory cytokines (IL-1β and IL-6 in BALF, IL-1β, TNF-α and IL-6 in LPA serum), markedly inhibited the phosphorylation of NF-kB p65. However, it inhibited the expression of MyD88 (myeloid differentiation factor 88) though not the expression of NF-κB p65 in lung tissues, which indicates that bergenin partially suppresses its production. In addition, bergenin also inhibited nuclear translocation and phosphorylation of NF-κB p65 stimulated by LPs in Raw264 cells, indicating that bergenin has anti-inflammatory effects in LPS-induced pulmonary edema [109]. 

Wistar rats were induced to acute ulcerative colitis by TNBS (2,4,6-trinitrobenzenesulfonic acid), causing an intestinal inflammation with tissue damage, and were subsequently treated with bergenin at concentrations of 12 to 100 mg/kg/day [137]. The results indicated that bergenin (IC_50_ of 25mg/kg) decreased the signs of macroscopic and microscopic colitis damage and reduced the degree of neutrophilic infiltration in the colon tissue. In addition, it was able to negatively regulate the expression of COX-2, iNOS, IkB-α, and pSTAT3 proteins (phosphorylated transducer and activator of immunohistochemical expression of expression-3) as well as activating inflammasome signaling pathways [137]. This pathway consists of an intracellular multiprotein complex that acts in the activation of enzymes of the cysteine-aspartate proteases (CASPASES) family as an essential structure for the regulation of immunity under physiological conditions and in recognition of danger signals with subsequent recruitment of cytokines that will normalize inflammation [138] The findings did not differ from those of Wang et al. [139] who tested bergenin isolated from the herb *S. stolonifera*. Gao et al. [10] reported that bergenin plays an anti-inflammatory role through the modulation of MARPK and NF-kB signaling pathways in an LP-induced mammary gland mastitis mouse model. Mice with mastitis induced in the mammary gland showed an excessive concentration of the pro-inflammatory cytokines NO, TNF-α, IL-1β, and IL-6, but only after treatment with bergenin. The results suggest that bergenin reduced the expression of pro-inflammatory cytokines, NO, TNF-α, IL-1β, and IL-6 by inhibiting the activation of the signaling pathways NF-kB and MAPKs, resulting in tissue normalization. These two pathways are responsible for the expression of inflammatory processes [140]. Inhibition of these two pathways was also observed in BALB/c rats (IC_50_ of 10 µM) with inflammation caused by *Klebsiella pneumonia* [141]. Souza et al. [142] reported that the phenolic extract of the stem bark of *E. uchi* (the main constituent being bergenin) had an anti-inflammatory effect in the edema of mice paws induced by intraplantar injection of carrageenan. Carrageenan is an inflammatory agent and produces inflammation by releasing prostaglandins, causing the formation of edema [143]. The results suggested that the phenolic extracts normalized edema by inhibiting COX-2, the main COX isoform induced during inflammation, to regulate the production of prostaglandins at the site of inflammation [128]. These results are similar to the findings of Borges [142], who tested acetylbergenin isolated from the stem bark of *E. uchi* in a paw edema model in rats induced experimentally by intraplantar injection of 100 µL of 1% carrageenan in the right paw of rats of the MacCoy lineage. Bergenin had an anti-inflammatory effect in a model of inflammation induced by Freund’s complete adjuvant (FCA) [73,144]. Mice were experimentally induced into edema by injection of FCA in the plantar region of the right paw. After this procedure, the recruitment of pro-inflammatory cytokines in the site was observed. The treatment with bergenin determined the suppression (IC_50_ of 12.5 mg/kg) of IL-1β, TNF-α, and IL-10 levels and normalized the tissues, suggesting that the anti-inflammatory effect of bergenin may be closely linked to the inhibition of these inflammatory cytokines [145]. It is known that they precede the release of the final mediators of hyperalgesia, i.e., prostaglandins and sympathetic amines [146]. These results are similar to the findings of Bharate et al. [22], who tested bergenin-rich extracts of *B. ciliata*.

Male Sprague-Dawley rats were experimentally exposed to tobacco smoke and developed chronic bronchitis. Later, they were treated with bergenin (87 mg/kg) and dexamethasone (0.2 mg/kg). The results indicated that both compounds suppressed inflammatory cell infiltration and inhibited mucus secretion, in addition to reducing white blood cells in BALF [147]. The authors suggest that the anti-inflammatory mechanism of action of bergenin may be associated with the alteration of branched-chain amino acid (BCAA) metabolism, glycine, serine, and threonine metabolism, and glycolysis to treat chronic bronchitis. It has been proven that these metabolic changes have an influence on the inflammatory response [148].

**Table 3 biomolecules-13-00403-t003:** Studies related to the anti-inflammatory activity of bergenin and its derivatives.

Compound Name	Source of Isolation	Type of Assay, IC_50_	Cytotoxicity		Reference
			CL	CV	
Bergenin	Trunk bark of *E. uchi*	in vitro, 1.2 µM	NC	NC	[131]
Bergenin	*Bergenia spp.*	in vivo -mice, 50 mg/kg	Raw264.7 cells	NC	[109]
Bergenin	NI	in vitro, 50 µM	CCK8Cells	NC	[134]
Bergenin	*Peltophorum spp.*	in vivo -rats, 25 mg/kg	ND	ND	[137]
Bergenin	NI	in vitro, 10 µM	Raw264.7 cells	NC	[141]
Bergenin	Trunk bark of *E. uchi*	in vivo -mice, 100 mg/kg	ND	ND	[143]
Bergenin	NI	in vivo -rats, 50 mg/kg	ND	ND	[138]
Bergenin	Bark, leaf, and branches of *E. uchi*	in vitro, 1.2 µM	ND	ND	[132]
Acetylbergenin	Stem bark of *E. uchi*	in vivo -rats, 6.8 mg/kg	ND	ND	[144]
Bergenin	Stem bark of *C. gardnerianum*	in vivo -mice,12.5 mg/kg	BALB/c mice splenocytes	NC	[73]
Bergenin	Crude extract of *M. philippenensis*	in vitro, 303.12 µM	NI	NC	[126]
Heptylbergenin	Crude extract of *M. philippenensis*	in vitro, 212.95 µM	NI	NC	[126]
Octylbergenin	Crude extract of *M. philippenensis*	in vitro, 269.99 µM	NI	NC	[126]
Ethylbergenin	Crude extract of *M. philippenensis*	in vitro, 322.09 µM	NI	NC	[126]
Propylbergenin	Crude extract of *M. philippenensis*	in vitro, 303.12 µM	NI	NC	[126]
Bergenin	*S. stolonifera*	in vivo- mice, 100 mg/kg	RAW264.7 cells	NC	[139]
Bergenin	*S. stolonifera* herb	in vitro, 0.1 µM	RAW264.7Cells	NC	[139]
Bergenin	NI	in vivo -rats, 87 mg/kg	ND	ND	[147]
Bergenin	NI	in vitro, 100 µM	CCK-8	NC	[135]
Bergenin	NI	in vitro, 30 µM	ND	ND	[136]
Bergenin	Stem bark of *C. gardnerianum*	in vivo -rats, 25 mg/kg	Macrophages	NC	[149]
Bergenin	Rhizome of *Bergenia spp.*	in vitro, 7.29 μM	INS-1E rat insulinoma cells	NC	[150]
Bergenin	Extract of *B. ciliata*	in vitro, 12.5 μg/mL	THP-1	NC	[22]
Bergenin	Extract of *B. ciliata*	in vivo -mice, 100 mg/kg	THP-1	NC	[22]
11-*O*-(40-*O*-Methylgalloyl)-bergenin	Methanolic extract of *S. atrata*	in vitro, 100 μg	ND	ND	[129]
Bergenin	Ethanolic extract of dry leaves of *T. procumbens*	in vitro, 70.54 µM	ND	ND	[66]
Bergenin	Ethanolic extract of dry leaves of *T. procumbens*	in vivo -rats, 200 mg/kg	ND	ND	[66]
11-*O*-Galloylbergenin	Ethanolic extract of *M. philippinensis*	in vivo -rats, 20 mg/kg	LCMK-2 monkey kid-ney epithelial cells and mice hepatocytes	NC	[151]
Acetylbergenin	Stem bark of *E. uchi*	in vivo -rats, 6.8 mg/kg	ND	ND	[152]

Abbreviation: NC: no cytotoxicity, ND: no cytotoxicity determined, NA: no activity, CL: cell line, CV: cell viability, NI: Uninformed.

### 5.8. Antioxidant Activity

Bergenin isolated from parts of different plant species has shown good antioxidant activity in 33 in vitro and 4 in vivo studies (Table 4). Bergenin has shown an in vitro effect on free radical scavenging in assays with the DPPH(2,2-diphenyl-1-picrylhydrazylhydrate) radical. The DPPH free radical method is an antioxidant assay based on electron transfer that yields a violet solution in ethanol [153]. The IC_50_ for free radical scavenging DPPH in vitro was determined as 0.7–165.35 µg/mL [49,66,73,109,124,125,126,127,128,129,130,131,132,133,134,135,136,137,138,139,140,141,142,143,144,145,146,147,148,149,150,151,152,153,154,155,156,157,158] and 951 µM [46]. The presence of bergenin reduces free radicals and gives rise to a colorless ethanol solution [159], indicating that it may be potentially useful for various pathological conditions associated with the devastating effects of oxygen-reactive species [124]. Studies with the same approach also evaluated the antioxidant potential using bergenin derivatives for DPPH free radical scavenging and presented a satisfactory IC_50_, i.e., 11-*O-*galloylbergenin IC_50_ of 5.39–7.45 µg/mL [105,108], hydroxybenzoyl-bergenin IC_50_ of 7.45 µg/mL [160], with the exception of diethyl ether of bergenin IC_50_ of 400 µg/mL [154]. Bergenin also exhibited antioxidant activity in vitro in hydrogen peroxide radical scavenging activity (H_2_O_2_) IC_50_ of 32.54 μg/mL [73]; in superoxide radical scavenging activity IC_50_ of 0.25–100 μg/mL [47,66,73]; in ABTS radical scavenging activity IC_50_ of 31.56–75.06 μg/mL [66,73] and 0.08 mg/mL [155]; in lipid peroxidation scavenging activity (LP) IC_50_ of 365.12 μg/mL [73]; in nitric oxide, radical scavenging activity (NO) IC_50_ of 2.98–785.63 μg/mL [73,155] and 0.35 mg/mL [47]; in hydroxyl radical scavenging activity (HO) IC_50_ of 0.12 mg/mL [48], 8.48 μg/mL [155]; in the radical NADH IC_50_ of 1 mg/g [156] and in the radical FRAP IC_50_ of 0.4 mg/g [156].

Bergenin exhibited antioxidant activity in HepG2 cells that were induced to oxidative damage by sodium selenite in vitro [161]. The cells treated with sodium selenium caused an oxidative-antioxidant imbalance that damaged them as it decreased antioxidant enzymes, which allowed the proliferation of free radicals that, in turn, triggered pro-inflammatory cascades and induced the damage. HepG2 was co-incubated with sodium selenite (10 µM) and bergenin (75, 150, and 300 µM) for 24 h. The results indicated that bergenin (IC_50_ of 75 µM) exerted protective effects against oxidative stress induced by sodium selenite in HepG2 cells since it sequesters free radicals and normalizes the oxidative stress balance. Similar protection was observed in PC12 cells, a cell line derived from a pheochromocytoma of the adrenal medulla of rats, with norbergenin derivatives isolated from *D. crassiflora* stems (IC_50_ of 0.1 and 10 µM) [162].

BALB/c mice immunosuppressed by cyclophosphamide (Cy) experimentally induced a decrease in the action of enzymes [superoxide dismutase (SOD), catalase (CAT), and glutathione peroxidase (GSH-Px)], which are important in response to oxidative stress and accelerate the formation of free radicals and, consequently, the spleen and thymus suffered lesions [163]. Manente et al. [164] reported that oxidative cell damage is a major side effect of chemotherapy drugs, including cyclophosphamide. Cyclophosphamide disrupts the redox balance and causes tissue damage; however, after treatment with bergenin, the results suggested that bergenin (IC_50_ of 20 mg/kg) reversed the Cy-induced decrease in total antioxidant capacity, including superoxide dismutase (SOD), catalase (CAT) and glutathione peroxidase (GSH-Px) activities. This improved humoral and cellular immune functions and increased antioxidant activity. This effect is justified by the fact that bergenin acts as a free radical scavenger or a redox-regulating agent, effectively avoiding Cy-induced oxidative stress injury by increasing antioxidant enzymes and oxidative-reducing enzymes [163]. These enzymes, when increased at the site of injury, convert harmful oxygen into less-reactive hydrogen peroxide, catalyzing the dismutation of superoxide into oxygen and hydrogen peroxide [164]. Lee et al. [165] reported that bergenin had an antioxidant effect in mice treated with morphine. Morphine treatment induced oxidative stress by decreasing antioxidant enzymes and causing the proliferation of free radicals, and this allowed inflammation of the brain since oxidative stress and inflammation are interdependent. Free radicals activate pro-inflammatory genes that trigger a cascade of progressive inflammation; however, treatment with bergenin (IC_50_ of 20 mg/kg) had an antioxidant effect and played a role in antinarcotic effects through adaptation to morphine-induced oxidative stress in the brain. Sriset et al. [162 reported that bergenin attenuates sodium selenite-induced hepatotoxicity by improving hepatic oxidant-antioxidant balance in ICR mice. Mice were orally administered sodium selenite (4 m/kg), which caused liver damage through oxidative-antioxidant imbalance. This is because there is an increase in plasma levels of enzymes aspartate aminotransferase, alanine aminotransferase, and alkaline phosphatase, and there is an increased proliferation of reactive oxygen species by decreasing antioxidant enzymes, resulting in lipid peroxidation in plasma. After treatment with bergenin, the results indicated that bergenin (IC_50_ of 10 mg) restored normal tissue damaged by the antagonistic effect exhibited by sodium selenite. The effect of bergenin on membrane lipid peroxidation and ascorbic acid level in tissues was studied using pathogen-free weaned rats as the experimental animal and 2,4-dinitrophenyl hydrazine (2,4-DNPH) as the experimental oxidant [166,167]. Lipid peroxidation was experimentally induced with 2,4-DNPH via the intraperitoneal route, which allowed a greater proliferation of reactive oxygen species and triggered processes that led to tissue damage by oxidative-antioxidant imbalance. In this context, three primary antioxidant enzymes were analyzed, namely catalase, superoxide dismutase (SOD), and glutathione peroxidase, and two non-enzymatic antioxidants, namely vitamin E (*α*-tocopherol) and vitamin C (ascorbic acid). After treatment with bergenin isolated from *S. gabonensis* stem bark extract (main constituent being bergenin), the results indicated that the bark extract (IC_50_ of 2.8 mg/100 g) exhibited divergent effects on antioxidant enzymes: impaired the enzyme-inducing action of 2,4-DNPH on liver and red blood cell catalase, reduced the depressant effect of SOD, and neither 2,4-DNPH nor the extract had any measurable effect on glutathione peroxidase. The bark extract also exerted a sparing effect on tissue antioxidant vitamins, ascorbic acid, and vitamin E, effectively inhibiting their depletion by 2,4-DNPH in the liver, red blood cells, and brain. These results suggest that the antioxidant mechanism of action of the bark extract against membrane peroxidation is multifactorial/multisystem, involving catalase inhibition, enhancing the SOD capacity of the liver and red blood cells, and sparing tissue depletion/use of vitamins C (ascorbic acid) and E (α-tocopherol) [166].

**Table 4 biomolecules-13-00403-t004:** Studies relating to the antioxidant activity of bergenin and its derivatives.

Compound Name	Source of Isolation	Type of Assay + IC_50_	Cytotoxicity		Reference
			CL	CV	
Bergenin	Roots of *B. ligulata*	in vitro/DPPH, 100 µg/mL	ND	ND	[105]
11-*O*-Galloylbergenin	Roots of *B. ligulata*	in vitro/DPPH, 7.45 µg/mL	ND	ND	[105]
Bergenin	Extracts of the aerial parts of *B. ligulata*	in vitro/DPPH, 54 μg/mL	ND	ND	[108]
Hydroxybenzoyl-bergenin	Extracts of the aerial parts of *B. ligulata*	in vitro/DPPH, 7.45 μg/mL	ND	ND	[108]
11-*O*-Galloylbergenin	Extracts of the aerial parts of *B. ligulata*	in vitro/DPPH, 5.39 μg/mL	ND	ND	[108]
Bergenin	Stem bark of *P. pterocarpum*	in vitro/DPPH, 0.96 μg/mL	ND	ND	[154]
Bergenin diethyl ether	Stem bark of *P. pterocarpum*	in vitro/DPPH, 400 μg/mL	ND	ND	[154]
Bergenin	Roots of *C. digyna*	in vitro/DPPH, 165.35 μg/mL	ND	ND	[73]
Bergenin	Flowers of *P. pterocarpum*	in vitro/DPPH, 1.95 μg/mL	ND	ND	[155]
Bergenin	Ethanol extract from dried leaves of *T. procumbens*	in vitro/DPPH, 20.42 μg/mL	ND	ND	[66]
Bergenin	Stem bark of *M. japonicus*	in vitro/DPPH, 951 μM	ND	ND	[46]
Bergenin	*M. philippensis*	in vitro/DPPH, 99.807 μg/mL	ND	ND	[49]
11-*O*-Galloylbergenin	*M. philippensis*	in vitro/DPPH, 7.276 μg/mL	ND	ND	[49]
Bergenin	Rhizome of *B. ciliata*	in vitro/DPPH, 100 μg/mL	ND	ND	[157]
Bergenin	*E. uchi*	in vitro/DPPH, 4.02 μg/mL	ND	ND	[158]
Bergenin	Rhizome of *B. ciliata*	in vitro/DPPH, 0.7 mg/g	ND	ND	[156]
Bergenin	Aerial parts of *T. labialis*	in vitro/DPPH, 100 μg/mL	ND	ND	[63]
Bergenin	Bark, leaf, and branches of *E. uchi*	in vitro/DPPH, 24.20 μg/mL	J774 cells of murine macrophages	NC	[168]
Bergenin	Ethanol extract from dried leaves of *T. procumbens*	in vitro/H_2_O_2_, 100 μg/mL	ND	ND	[66]
Bergenin	Roots of *C. digyna*	in vitro/H_2_O_2_, 32.54 μg/mL	ND	ND	[73]
Bergenin	Roots of *C. digyna*	in vitro/H_2_O_2_, 75.06 μg/mL	ND	ND	[73]
Bergenin	*M. repandus*	in vitro/H_2_O_2_, 0.25 mg/mL	ND	ND	[47]
Bergenin	Roots of *C. digyna*	in vitro/ABTS, 75.06 μg/mL	ND	ND	[73]
Bergenin	Ethanol extract from dried leaves of *T. procumbens*	in vitro/ABTS, 31.56 μg/mL	ND	ND	[66]
Bergenin	*M. repandus*	in vitro/ABTS, 0.08 mg/mL	ND	ND	[47]
Bergenin	*M. repandus*	in vitro/OH, 0.12 mg/mL	ND	ND	[47]
Bergenin	Flowers of *P. pterocarpum*	in vitro/OH, 8.48 μg/mL	ND	ND	[155]
Bergenin	Rhizome of *B. crassifolia*	in vitro/FRAP, 0.4 mg/g	ND	ND	[156]
Bergenin	Rhizome of *B. ornata*	in vitro/NADH, 1 mg/g	ND	ND	[156]
Bergenin	Roots of *C. digyna*	in vitro/NO, 785.63 μg/mL	ND	ND	[73]
Bergenin	*M. repandus*	in vitro/NO, 0.32 mg/mL	ND	ND	[47]
Bergenin	Flowers of *P. pterocarpum*	in vitro/NO, 2.98 μg/mL	ND	ND	[155]
Bergenin	Roots of *C. digyna*	in vitro/PL, 365.12 μg/mL	ND	ND	[73]
Bergenin	NI	in vitro, 75 μM	HepG2	NC	[161]
Bergenin	NI	in vivo -mice, 20 mg/kg	Splenic NK and CTL cells	NC	[163]
Bergenin	NI	in vivo -mice, 20 mg/kg	NI	NI	[165]
Bergenin	NI	in vivo -mice, 10 mg/kg	HepG2	NC	[161]
Bergenin	Extract of stem bark from *S. gabonensis*	in vivo -mice, 2.8 mg/100 g	ND	ND	[166]
4-*O*-*p*-Hydroxybenzoylnorbergenin	Leaves of *D. gilletii*	in vitro/DPPH, 8.2 μg/mL	ND	ND	[169]
Methylbergenin	Whole plant of *A. japonica*	in vitro/NO, 38.4 μg/mL	HepG-2	NC	[170]

Abbreviation: NC: no cytotoxicity, ND: no cytotoxicity determined, NA: no activity, CL: cell line, CV: cell viability, NI: Uninformed.

### 5.9. Antinociceptive Activity

The antinociceptive activity is linked to the analgesic activity of bergenin, blocks the sensory neurons, and gives the sensation of pain relief in vivo. Obviously, all the studies presented in the previous sections, especially the anti-inflammatory and antioxidant activities, show the ability of bergenin to restore the normality of inflamed tissue or an organ or to restore the antioxidant oxidant balance and cause the mice or rats to feel pain relief. Ethanolic extract of *D. dentatus* leaves, which are rich in bergenin, showed antinociceptive activity in male Swiss mice. The mice were induced by formalin intraperitoneally, causing joint inflammation, knee edema, leukocyte infiltration, hyperalgesia, and sensation of excessive cold; however, oral treatment with the extract (100 and 300 mg/kg) reduced leukocyte infiltration and normalized the injured tissues, which consequently significantly inhibited nociceptive sensitivity and sensitivity to cold (IC_50_ of 100 mg/kg). The results suggest that the ethanolic extract could interfere with the peripheral and central pain mechanisms of nociception induced by formalin [171]. Bharate et al. [22] reached the same conclusion using the ethanolic extract of the rhizome of bergenin-rich *B. ciliate* (IC_50_ of 2 g/kg) in rats induced to nociception by acetic acid and the formalin-induced paw-licking method and suggested that upregulation of inflammation is inversely proportional to antinociceptive activity, i.e., the bergenin contained in the extract enables inhibition of both nociceptive response phases as anti-inflammatory processes are activated. Souza et al. [143] reported that the extract of the bark of *E. uchi* presented good antinociceptive activity (IC_50_ of 100 mg/kg) in the test of abdominal contortions induced by acetic acid. They suggested that the mechanism of action is associated with the inhibition of the formation of pro-inflammatory mediators. Paulino et al. [62] claim that bergenin has the potential to exert antinociceptive activity and may affect the opioid system. Opioids act by promoting the opening of potassium channels and inhibiting the opening of voltage-gated calcium channels, thus causing hyperpolarization and reduced neuronal excitability. In addition, there is a reduction in the release of transmitters (by inhibiting the entry of Ca^2+^), blocking the sensation of pain. 

Bergenin isolated from the rhizome of *B. ciliata* showed antinociceptive activity (IC_50_ of 10 mg/kg) by normalizing the antioxidant oxidative balance and de-accumulation of IL-2 in an induced hyperoxaluric rat model [172]. Semisynthetic acetylbergenin, the product of acetylation of bergenin isolated from the cortex of *E. uchi*, also showed good antinociceptive activity in Swiss albino rats with abdominal contraction [173]. The abdominal contraction was induced in rats by intraperitoneal injection with 0.6% acetic acid triggering contractions, thereby raising levels of prostaglandin E2, which triggers local inflammatory processes by raising COX-2, in addition to interfering with nociceptive mechanisms causing extreme pain in rats; however, treatment with bergenin (IC_50_ of 6.8 mg/kg) normalized inflammation and inhibited nociception induced by acetic acid.

### 5.10. AntiArthritic Activity

Bergenin has been studied in order to verify its antiarthritic potential. In rat or mouse models that experimentally induced inflammation in the joints, seven studies were able to discriminate antiarthritic activity in vivo. Branquinho et al. [171] reported the activity of the ethanolic extract of leaves of *D. dentatus*, which are rich in bergenin, in male Swiss mice against rheumatoid arthritis. In the study, mice were experimentally induced to joint inflammation (including knee edema, leukocyte infiltration, and hyperalgesia) by zymosan through a single intraperitoneal dose of 200 µL. After treatment with the extract (0.3 to 30 mg/kg), the results suggested that the extract showed therapeutic action in rheumatoid arthritis and decreased the amount of inflammation through the de-accumulation of pro-inflammatory cytokines and by modulating the immune response in the cells. Bharat et al. [122] came to the same conclusion using the ethanolic extract of the rhizome of bergenin-rich *B. ciliata* (IC_50_ of 2 g/kg) in rats induced to arthritis using *Mycobacterium*. 

Bergenin isolated from the rhizome of *B. ciliata* was shown to be a potent antiarthritic agent in an induced hyperoxaluric rat model [172]. Hyperoxaluria was induced using ethylene glycol, and this induction caused oxidative stress, especially in the joints, which caused a proliferation of pro-inflammatory cytokines, causing edema. However, treatment with bergenin (IC_50_ of 10 mg/kg) inhibited the proliferation of IL-β and sequestered reactive oxygen species, thus normalizing the joints. In the study by Jain et al. [174], bergenin and its derivatives exerted antiarthritic activity, possibly by inhibiting pro-inflammatory cytokines and producing TNF-α. Nazir et al. [175] reported that the effect of bergenin and norbergenin against adjuvant-induced arthritis, both at doses above 2 mg/kg, is achieved by possible modulation of the Th1/Th2 cytokine balance. 

Methanolic extract of *C. capitella* and its metabolite 11-*O-*(4′-*O-*methyl galloyl)-bergenin, isolated from aerial parts, showed an antiarthritic effect on rat hind paws. Arthritis was induced by intraplantar injection of 0.1 mL of complete Freund’s adjuvant (CFA) into the subplantar tissue of the hind paw. CFA produced definitive edema within 24 h, with progressive arthritis on day 9 after inoculation according to the elevated levels of TNF-α, IL-1β, IL-6, and myeloperoxidase activity. There was also an increase in rheumatoid factor (RF) and anticyclic citrullinated peptide antibody (antiCCP), which are considered predictors of severe arthritis. Subsequently, they were treated with the extract (250 mg/kg) and 11-*O-*(4′-*O-*methyl galloyl)-bergenin (5, 10, and 20 mg/kg). The findings indicated that both treatments inhibited the production of pro-inflammatory cytokines, and serum levels of RF and antiCCP were normalized, with an IC_50_ of 250 mg/kg for the extract and 20 mg/kg for 11-*O-*(4′-*O-*methyl galloyl)-bergenin [176].

### 5.11. Antiulcerogenic Activity

The antiulcerogenic activity of bergenin has been reported in several experimental ulcers. In this context, about six studies were carried out, one in vitro and five in vivo. Bergenin and norbergenin (isolated from the aqueous extract of *M. japonicus*) showed an inhibitory effect on the bovine adrenal tyrosine hydroxylase (TH) enzyme in vitro [177]. The increased activity of this enzyme is suggested to be related to the occurrence of gastric ulcers in patients stressed by the cold. The findings showed that bergenin and norbergenin inhibited TH activity by 12.2% and 51.0% at the concentration of 5 µg/mL, 16.1% and 51.6% at the concentration of 10 µg/mL, and 29.0% and 53.4% at the concentration of 20 µg/mL, respectively [177].

Male Donryu rats received treatment with bergenin (30 to 1000 mg/kg) orally and were subsequently placed in a cage and immersed up to the level of the xiphoid process in a water bath (23 °C) for 7 h, then euthanized. The stomach was removed and submerged in a formalin solution inducing inflammation, allowing the marking of the inner and outer layers of the gastric wall. Finally, the stomach was incised along the curvature, and the lesions were examined [178]. The results indicated that bergenin (IC_50_ 30 mg/kg) inhibited the development of stress-induced gastric ulcers in rats. One of the mechanisms attributed to its effectiveness may be the inhibition of acetylcholine release, which induces acid secretion and improves gastric motility [178]. Oral administration of bergenin and norbergenin isolated from the leaves and roots of *F. microcarpa* and luvangetin showed significant protection against pyloric gastric ulcers in rats induced by aspirin and colds, suggesting that the gastroprotective effects of bergenin and norbergenin may be due to increased prostaglandin production [41].

 Semisynthetic acetylbergenin (1i) isolated from *E. uchi* was effective in preventing gastric ulcers [152]. Acute gastric ulcer was induced by stress, according to Basile et al. [41]. Wistar rats were fasted with free access to water for 24 h, then treated with distilled water (0.5 mL), acetylbergenin (6.8 mg/kg), and indomethacin (10 mg/kg). Each animal was kept for 17 h in a container tube, which was immersed vertically until the water reached the neck region of the animal in a tank with running water at 25 °C. In addition, the rats were euthanized by CO_2_ inhalation. Their stomachs were immediately excised and opened by cutting along the largest curvature for examination of the inner wall, and the lesions were counted. The results indicated that treatment with indomethacin (10 mg/kg) produced more lesions when compared with acetylbergenin at a dose of 6.8 mg/kg, revealing that acetylbergenin has a protective effect [152].

### 5.12. Antidiabetic/Antiobesity Activity

Antidiabetic/antiobesity activity was investigated in eight in vivo studies. The alcoholic extract of *C. digyna* root rich in bergenin showed an antidiabetic effect in diabetic rats induced by streptozotocin-nicotinamide. The rats received a single dose of streptozotocin-nicotinamide (65 mg/kg) intraperitoneally, which induced fatal hypoglycemia due to the massive release of pancreatic insulin. After 6 h, the rats received 10% glucose solution and continued receiving more glucose by injection until reaching a glucose level of 200 mg/dL in fasting. Treatment with the extract (250 to 750 mg/kg) for 14 days reduced dose-dependent blood glucose levels from 200 mg/dL to 146.33 mg/dL (IC_50_ of 250 mg/kg), and it was also noted that it was able to reduce the body weight of the rats [179]. Hyperglycemia is recognized as a common complication of diabetes mellitus. Reduced insulin secretion causes a variety of disruptions in metabolic and regulatory mechanisms that lead to lipid accumulation. Bergenin significantly reduces triacylglycerols and total cholesterol in diabetic rats. The lipid-lowering effect of bergenin and other antidiabetic drugs reduces the risk of vascular complications [179]. San et al. [91] reported that the methanolic extract of the root of *C. javana* affected glucose uptake (100 µg/mL) in L6 skeletal muscle cells according to the reported methods [91,180].

Rats fed a hyperglycemic diet increased blood glucose to 220 mg/dL and developed type 2 diabetes, then received intragastric treatment with bergenin (10, 20, 40 mg/kg) or metformin (25 mg/kg) as the positive control from 8 to 16 weeks [181]. Treatment with bergenin (10, 20, 40 mg/kg) significantly reduced the concentration of glucose in the blood from 220 to 86.19 mg/dL against 80.45 mg/dL of metformin. Bergenin significantly improved the insulin sensitivity index, reduced liver damage and oxidative changes, and brought antioxidants and lipids back to normal, suggesting that bergenin can be used as a functional drug or as an adjunct in the management of insulin resistance and associated fatty liver disease [181]. Other studies have also determined the IC_50_ for the inhibition of the development of type 2 diabetes with antiobesity consequences; namely, bergenin isolated from the methanolic extract of *F. racemosa* (IC_50_ of 200 mg/kg) [182] and bergenin isolated from the rhizomes of *B. crassifolia* (IC_50_ of 100 mg/kg) [183].

Methoxybergenin, a natural derivative isolated from the stem bark of *V. pauciflora*, showed an antidiabetic effect in diabetic Wistar rats induced via alloxan [184]. Twenty-five Wistar rats with a body weight of 100 g, presenting a normal glycemia (50–125 mg/dl), were induced via alloxan at a dose of 150 mg/kg intraperitoneally, which subsequently presented an elevation of the blood glucose level (440 mg/dL), thus damaging the production of insulin by the cells of the pancreas. Treatment with methoxybergenin after 21 days of treatment exhibited good antidiabetic activity (IC_50_ of 191 mg/200 g), and this inhibition allowed the reduction of the body weight of the rats, which suggests antiobesity activity [184].

Kumar et al. [185] reported that bergenin isolated from the methanolic extract of *M. philippinensis* showed moderate antiglycation activity (IC_50_ of 186.73 µg/mL), which suggests participation in glycemic regulation. This effect allows sugar molecules not to be fixed in large quantities in proteins, especially hemoglobin, in addition to avoiding increased oxidative stress. The same was observed with respect to 11-*O-*galloylbergenin isolated from the ethyl acetate fraction of *P. peltatum* (IC_50_ of 0.1 µg/mL) [186].

### 5.13. AntiArrhythmic Activity

Studies on the antiarrhythmic activity of bergenin are scarce; however, only one study undertook this assessment [7]. Bergenin isolated from the aerial parts of *F. virosa* was investigated for its antiarrhythmic effects at concentrations of 0.2 mg/kg, 0.4 mg/kg, and 0.8 mg/kg, and showed distinct therapeutic effects on arrhythmias induced by barium chloride (BaCl2) in rats. At concentrations of 0.4 mg/kg and 0.8 mg/kg, bergenin significantly countered arrhythmias induced by coronary artery ligation and reperfusion. At a concentration of 0.8 mg/kg, bergenin raised the atrial fibrillation threshold in rabbits from 1.34 mV to 1.92 mV, suggesting that bergenin has the potential to treat cardiac arrhythmias [7]. According to Filho et al. [187], the elevation of the atrial fibrillation threshold allows the electric current to travel with sufficient intensity to contract the muscles of the ventricles and cause involuntary systole. 

### 5.14. Anticancer Activity

The anticancer activity of bergenin has been reported in 13 in vitro studies and 2 in vivo studies. Bergenin-rich *E. agallocha* leaf extract showed anticancer activity in a cervical cancer cell line (SiHa HPV 16+) [45]. The extract was matched with a cancer cell line and peripheral healthy blood mononuclear cells (PBMC) using an MTT assay (3-(4,5-dimethylthiazol-2-yl)-2,5-diphenyltetrazolium bromide) to assess its cytotoxicity. The findings indicated that bergenin-rich extracts exhibited remarkable activity against the SiHa cell line (IC_50_ of 15.538 µg/mL), whereas for healthy cells (PBMC), the extracts induced cell proliferation, suggesting that they do not produce cytotoxicity in normal healthy cells. The mechanism of action of this effect may be associated with blocking the action of the GLI-related protein (glioma-associated oncogene homolog). GLI is a transcriptional effector involved in developing tumors; thus, blocking its action leads to the inhibition of its translocation to the nucleus [188]. Konoshima et al. [160] describe GLI as an effective inhibitor of the Hedgehog, a signaling pathway in cancer therapy. Bergenin has shown effects on human colorectal adenocarcinoma cell line HCT116 [189]. Bergenin markedly inhibited (IC_50_ of 30 µM) the growth of HCT116 cells with cellular apoptosis of up to 5.21%. When the influence of bergenin on the cell cycle was evaluated, it was found that it increased the number of cells in the Gap 1 (G1) phase with a decrease in the percentage of cells in the synthesis phase (S) in 24 h, showing that bergenin can lead to DNA damage in HCT116 cells by increasing the phosphorylation of the histone variant H2AX in Ser139 and generation of reactive oxygen species (ROS). The canonical PI3K/AKT/mTOR signaling pathway emerges as a critical regulator of the cell proliferation pathway [190]. This pathway is very important and is involved in the regulation of cell proliferation, cell cycle progression, apoptosis, and morphogenesis in different organs [191,192]. Bergenin, isolated from the bark of *F. religiosa*, *F. virens*, *F. glomerata*, *F. benghalensis*, and *T. pulpunea*, showed effects on cervical cancer lines of Hela and SiHa cancer cells (IC_50_ of 125.8 and 96.0 µg/mL respectively) using an MTT assay [42]. Newell et al. [77] reported anticancer activity of bergenin isolated from the seed of five *Ardisia* species, namely *A. japonica*, *A. escallonioides*, *A. mamillata*, *A. crenata,* and *A. compressa*, with the inhibition of human topoisomerase II enzyme and cytotoxicity in human liver cancer cells (HepG2) in vitro. The findings indicated that bergenin showed catalytic inhibition of topoisomerase II (IC_50_ of 18 µM) and cytotoxicity (IC_50_ of 18 µM) against HepG2 cells, and it was found that bergenin showed a tendency to accumulate cells in the G1 phase and reduction of G2/M leading to apoptosis of malignant cells. 

Bergenin (isolated from the herb *B. purpurascens*) inhibited the activity of human hepatic cytochrome P450 enzymes (CYP), CYP3A4, 2E1, and 2C9, with IC_50_ values of 14.39, 22.83, and 15.11 µM, respectively [193]. Enzyme kinetics studies showed that bergenin was not only a non-competitive inhibitor of CYP3A4 but also a competitive inhibitor of CYP2E1 and CYP2C9. These enzymes at high levels may favor extrinsic factors (e.g., alcohol and tobacco consumption), triggering the development of oral cancer [193].

Bulugahapitiya et al. [40] reported anticancer activity of ethyl acetate extract (Et**O**Ac) from *F. leucopyrus* leaves, bergenin and bergenin diastereomer (at C-9 and C-14) both isolated from Et**O**Ac, against human ovarian carcinoma. Anticancer activity was evaluated using cell proliferation assays (MTS) and human telomerase reverse transcriptase (hTERT) in human ovarian carcinoma (A 2780). The MTS assay showed significant antiproliferation activity with an IC_50_ of 36.35, 12.36, and 48.53 µg/mL for Et**O**Ac, bergenin, and bergenin diastereoisomer, respectively. A rapid depletion of the hTERT content in human ovarian cancer cells was observed for the bergenin diastereomer in the concentration range of 50–200 µg/mL. The findings suggested the anticancer activity of *F. leucopyrus* leaves against human ovarian cancer, and bergenin and its isomer were identified as the compounds responsible for the anticancer activity of this plant. It can effectively inhibit the proliferation of cancer cells and inhibit the human telomerase reverse transcriptase enzyme, which is directly responsible for the activation of cancer cell telomerase [194]. These results resemble the findings of Mettihewa et al. [195]. Bergenin, isolated from *S. stolonifera*, was shown to induce apoptosis (IC_50_ of 6.1 µM) in BGC-823 carcinoma cells in an MTT assay [196]. 

Esterified bergenin 5c (11-hydroxyl-modified bergenin), a semisynthetic derivative of bergenin, was evaluated for its antitumor properties in vitro and in vivo [197]. The findings showed that it trapped HepG2 cells (IC_50_ of 4.23 ± 0.79 mM) in the G2/M phase and induced cell apoptosis. In addition, compound 5c suppresses (IC_50_ 30 mg/kg) tumor growth in Heps xenograft-bearing mice with low toxicity. Liu et al. [198] reported that the methanolic extract, which is rich in bergenin and norbergenin, from *S. stolonifera* showed an antitumor effect, and its lung tumor inhibition rate can reach 49.2% (IC_50_ of 5.150 mg/g). The extract showed some capacity in the recovery of the immune system and hematological system of Lewis mice, in addition to causing necrosis in tumor cells and decreased macrophage density to inhibit the growth of lung tumors [198]. De-Biao et al. [199] reported anticancer activity of 3, 4, 11-trihydroxyl modified bergenin derivatives in an MTT assay in the inhibition of DU-145 and BGC-823 cells, with significant inhibitions of compounds 3a to 7a (3, 4, 11-trihydroxyl modified derivatives of bergenin). IC_50_ ranged from 20.89–100 µM for DU-145 and 23.99–100 µM. 

11-*O-*Galloylbergenin, a natural bergenin derivative isolated from leaves of *C. coreana*, was shown to be a potent antitumor agent in human osteosarcoma cells (MG63 cells) in an MTT assay. The findings showed that it inhibited the proliferation of MG63 cells and induced cellular apoptosis. This phenomenon was accompanied by the upregulation of the p53 and p21 genes [200]. The p53 gene encodes the tumor suppressor protein p53, which plays a significant role in tumor development and regulates the cell cycle and apoptosis, especially in the early events of osteosarcoma tumorigenesis [200].

### 5.15. Hepatoprotective Activity

In three in vitro studies and seven in vivo studies, bergenin showed hepatoprotective activity. Bergenin isolated from the cortex of *M. japonicus* exhibited good hepatoprotective activity by removing hepatotoxicity in a rat liver-cell assay induced by carbon tetrachloride (CCl_4_) [201]. Hepatocytes were isolated from rats using the method of Berry and Friend [202] and were then cultured [198]. After one day of plating, they were exposed to CCl_4_ (10 mM), which induced hepatotoxicity by metabolic activation. In other words, CCl_4_ is metabolically activated by cytochrome P-450-dependent mixed oxidase in the endoplasmic reticulum to form a trichloromethyl free radical (CCl_3_), which, combined with lipids and proteins in the presence of oxygen to induce lipid peroxidation [203], results in changes in the structures of the endoplasmic reticulum system and other membranes, loss of metabolic enzyme activation, reduced protein synthesis, and loss of glucose-6-phosphatase activation, thus leading to liver damage [204]. However, treatment with bergenin (IC_50_ of 300 μM) lowered the activity of enzymes whose high levels indicate liver damage, namely pyruvic transaminase (GPT) sorbitol dehydrogenase (SDH) [201]; both enzymes are also associated with lipid peroxidation [205]. In vivo studies positively confirmed the hepatoprotective activity in rats with experimentally induced hepatotoxicity by CCl_4_, and the IC_50_ was determined. Bergenin isolated from the cortex of *M. japonicus* exhibited a hepatoprotective effect in rats (IC_50_ of 50 mg/kg) and normalized the decreased activities of glutathione S-transferase and glutathione reductase, in addition to significantly preventing the elevation of hepatic malondialdehyde formation and depletion of reduced glutathione content in the liver [206]. Bergenin isolated from the herb *S. stolonifera* exhibited (IC_50_ of 100 mg/kg) a hepatoprotective effect and detoxified liver cells [206]; bergenin isolated from the extract of *P. pterocarpum* showed a hepatoprotective effect in albino rats (IC_50_ of 100 mg/kg) reducing excessive levels of the enzymes alanine aminotransferase (ALT), alkaline phosphatase (ALP), gamma-glutamyltransferase (γ-GT), direct bilirubin (DB) and total bilirubin (TB) [207,208]; bergenin and 11-*O-*galloylbergenin, isolated from the leaves of *A. edulis*, have remarkable antihepatotoxic activity against CCl_4_ and galactosamine cytotoxicity in cultured primary rat hepatocytes (IC_50_ of 100 mg/kg) [122]. Mondal et al. [209] reported that the hepatoprotective capacity of bergenin could also be related to the fact that it can sequester free radicals and restore the antioxidant oxidant balance.

The hepatoprotective effects of acetylbergenin have been examined against D-galactosamine (GalN). Ga1N induced liver damage in rats, compared to previously reported bergenin [137]. Acetylbergenin was synthesized from the acetylation of bergenin, isolated from *M. japonicus*, to increase lipophilic and physiological activities. Acetylbergenin was administered orally once daily for 7 days, and then GalN (400 mg/kg) was injected at 24 h and 96 h after the final acetylbergenin administration. Acetylbergenin reduced elevated serum enzymatic activities of alanine/aspartate aminotransferase, sorbitol dehydrogenase, glutamyltransferase, and GalN-induced hepatic malondialdehyde formation. Acetylbergenin also significantly restored GalN-induced decreased glutathione levels and decreased glutathione S-transferase and glutathione reductase activities to normalization. Therefore, these results suggest that acetylbergenin has hepatoprotective effects against GalN-induced hepatotoxicity by inhibiting lipid peroxidation and maintaining an adequate level of GSH for xenobiotic detoxification as its underlying hepatoprotective mechanisms. In addition, lipophilic acetylbergenin showed more activity in hepatoprotection than the much less lipophilic bergenin that was previously reported [210].

### 5.16. Neuroprotective Activity

The neuroprotective activity of bergenin has been reported in two in vitro studies. Bergenin showed neuroprotective activity against Alzheimer’s disease (AD) [211], a chronic progressive neurodegenerative disease, which often occurs in the elderly and negatively affected intellectual abilities and cognitive processes. Human neuroblastoma cell lines (SH-SY5Y) were treated in an MTT assay with *N*-methyl *D*-aspartate (NMDA) at the concentration of 2.5 mM per 24 h, which led to reduced cell viability (49.33%). In this context, this concentration was selected to induce cytotoxicity in SH-SY5Y, along with pretreatment with bergenin (5 to 50,000 nM) in an MTT assay. The findings revealed that pretreatment with bergenin led to NMDA concentration-dependent reversal in the concentration range of 5–500 nM. Bergenin at 500 nM led to the greatest increase in cell survival—up to 81.754% [211]. Takahashi et al. [46] reported the neuroprotective activity of norbergenin derivatives isolated from the methanolic extract of the bark of *M. japonicus*, and this suggests that it is closely related to their ability to sequester reactive oxygen species and thus restore antioxidant oxidative balance.

Suzuki et al. [212] reported the protective effect of norbergenin-11-caproate (a semisynthetic derivative of norbergenin isolated from *M. japonicus*) against cell damage in human neuroblastoma IMR-32 cells treated with tunicamycin. When IMR-32 cells were treated with tunicamycin, their viability in an MTT assay was decreased in a dose-dependent manner (0.01–1 μM). Treatment with norbergenin-11-caproate (10 mM) showed complete protection against the cell growth inhibitory effect of tunicamycin but did not inhibit the induction of Bip/GRP78 mRNA, suggesting the therapeutic potential of this derivative.

### 5.17. Cardioprotective Activity

Studies on the cardioprotective activity of bergenin are scarce, with only one study investigating this effect. Thirty rats were experimentally induced to myocardial infarction by isoproterenol (ISO), which sharply increased the ST and deep Q wave, in addition to causing leakage of cardiac marker enzymes, such as cTnI (cardiac troponin I), CPK (creatine phosphokinase), CK-MB (creatine kinase MB isoenzyme), LDH (lactate dehydrogenase), ALT (alanine aminotransferase) and AST (aspartate aminotransferase), from cardiac tissue to circulation, cell membrane rupture, hypoxia, and cardiac hypertrophy. Subsequently, they were treated twice within 24 h with bergenin at doses of 1 and 3 mg/kg via injection for 5 days to determine the cardioprotective response. The results indicated that bergenin (IC_50_ of 1 mg/kg) sharply restricted ST segment elevation induced by isoproterenol, Q wave, and ECG pattern, indicating that it has protective effects on the cell membrane since it prevents the extension of myocardial damage (induced by ISO) by strengthening the myocardial cell membrane and tissue architecture, and also normalizes marker enzymes [211].

## 6. Conclusions and Future Perspective

Because it is a secondary metabolite with immeasurable pharmacological potential and because it is distributed in many plant species (at least 112 species belonging to 34 families), bergenin has aroused the interest of researchers in the medical and biotechnological fields. Both its derivatives (natural and semisynthetic) and its extracts are well studied, with phytochemical confirmation of its highest concentration, and in none of the studies has it been observed to be cytotoxic to healthy cells. At least 17 activities were well studied, of which we can highlight the following: antimalarial, antileishmanial, trypanocidal, antiviral, antibacterial, anti-inflammatory, antioxidant, antinociceptive, antiarthritic, antiulcerogenic, antidiabetic, anticancer and hepatoprotective activities. However, it is emphasized that more studies should be carried out in order to further explore its pharmacological potential, especially the unraveling of the different mechanisms of action against infectious-contagious pathogens that are part of tropical and neglected infectious diseases and which have wide worldwide distribution such as malaria and leishmaniasis. For activities whose mechanisms of action are already well known, such as anti-inflammatory and antioxidant activities, there may be interest in the production of synthetic derivatives, with conducting of preclinical and clinical studies, since in preclinical studies in vitro and in vivo (in mice) with natural bergenin as well as its semisynthetic derivatives, it has been shown to be safe.

## Figures and Tables

**Figure 1 biomolecules-13-00403-f001:**
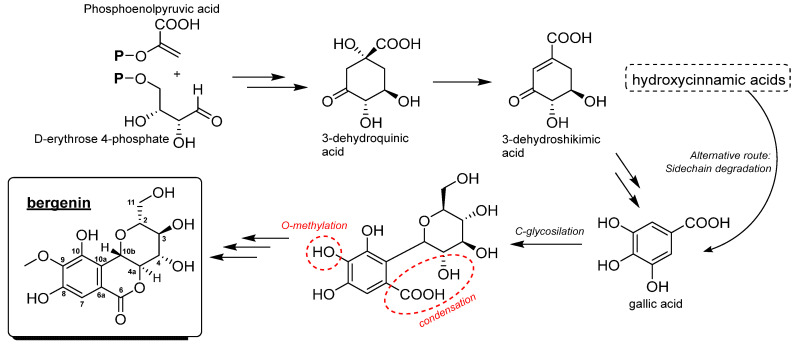
The biosynthesis of bergenin in plants is related to the gallic acid biosynthetic pathway.

**Figure 2 biomolecules-13-00403-f002:**
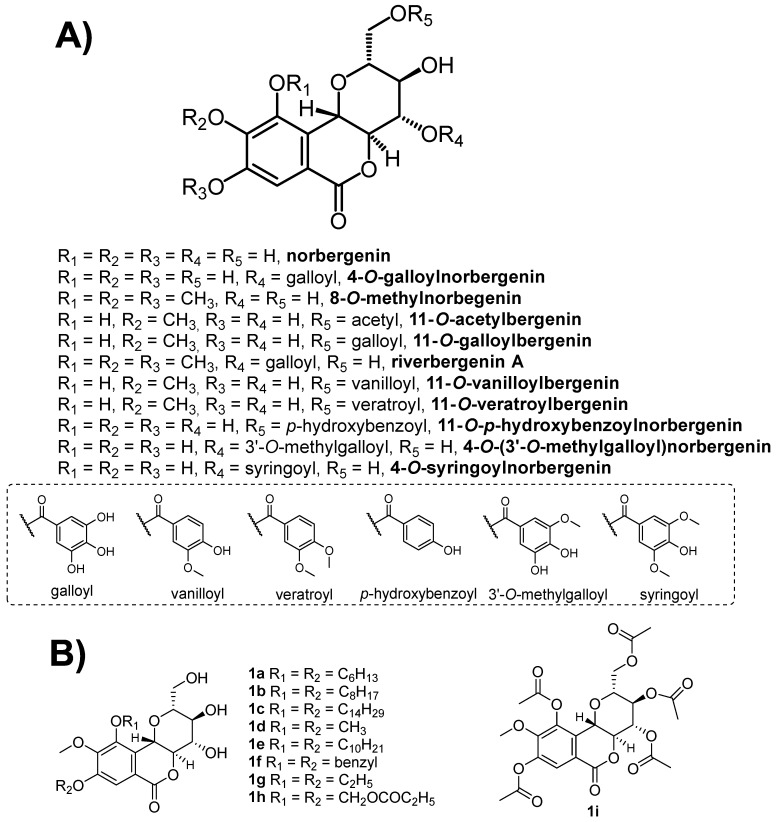
(**A**) natural derivatives of bergenin isolated from different plant species and (**B**) semisynthetic derivatives obtained by modulation of bergenin in the laboratory.

## Data Availability

No new data were created or analyzed in this study. Data sharing is not applicable to this article.

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
