# Peer review of "Chemistry and Pharmacology of Bergenin or Its Derivatives: A Promising Molecule"

_biomolecules, 2023, doi:10.3390/biom13030403_

Round 1
Author Response
Reply letter to reviewer 1
We address our greetings and gratitude to reviewer 1 for the rich suggestions, which we accept and correct,. In the paper, corrections suggested by reviewer 1 are highlighted in yellow.

Reviewer 2 Report
In the current review the authors described the sources for isolation of bergenin and its in vitro / in vivo biological and pharmacological activities. They underlined the fact that at least 17 biologic activities were well studied and that bergenin and both its natural and semisynthetic derivatives are active and nontoxic on healty cells. In my opinion, the review is interesting and well organized.
Some comments/suggestions:
Pg 2, Data collection:
-You wrote that you selected 230 papers, 2 master’s dissertations and 2 doctoral theses to write the review. You have 239 references. Please clarify where the difference comes from.
-line 81: what do you mean by “papers that had been withdrawn”.
Pg 8, Chemical aspects of bergenin:
- Considering the fact that bergenin has low solubility in water add please in which solvents (and in what volumes) can bergenin be solubilized in order to be able to study the biological activity.- Give please details concerning bergenin storage conditions and its stability (line 108)
Antimalarial activity:
-Pg 11, lines 201-202 and Table 2: please check if the mice were treated with 800mg/kg/day. Seems like a big dose for a mouse. -Table 2: for in vivo studies add please which studies were conducted on mice and which on rats
Antiviral activity:
Pg 15, lines 310-11: you wrote: “To date, about in vitro studies have been performed, though no study evaluated in vivo conditions”. You forgot to add the number of in vitro studies. Please add
Antibacterial activity:
Line 384, you added informations concerning Figure 3 but you forgot to put the figure 3 in you article. Please add
Anti-inflammatory activity - Concerning Shah et al study (lines 470-477):
-Which are the 16 different synthetic derivatives? Please add them
-I don’t understand which is the connection between what is presented at lines 470-77 and Figure 2. Please clarify.
Antioxidant activity:
Table 4: for in vivo studies add please which studies were conducted on mice and which on rats and what type of assays where performed - Tables 3 and 4 are not properly positioned in the article. Anticancer activity: - In my opinion is better to number the anticancer activity with 5.13 and to move the anti-arrhythmic activity before the cardioprotective activity - lines 856-868: compounds 5c, 3a to 7a are not presented in the article. Please clarify. Neuroprotective activity: Please add how many studies you found (in vivo and in vitro)Author Response
Reply letter to reviewer 2
We address our greetings and gratitude to reviewer 2 for the rich suggestions, which we accept and correct. In the paper, corrections suggested by reviewer 2 are highlighted in green.

Reviewer 3 Report
This review article deals with the chemistry and pharmacology of bergenin. It will add more information concerning this point in one article. It suitable for publication after corrections of the comments in the attached file.

Author Response
Reply letter to reviewer 3
We address our greetings and gratitude to reviewer 3 for the rich suggestions, which we accept and correct. In the paper, corrections suggested by reviewer 3 are highlighted in blue.

Reviewer 4 Report
The review "Chemistry and pharmacology of bergenin: a promising molecule " is very interesting, and the paragraphs have been well differentiated, in describing the different biological activities of the molecule. I ask the authors to insert a figure of the medicinal plant Bergenia crassifolia, from which bergenin was first isolated. Moreover, paragraph 5.14 should be reviewed carefully, since in many works, described and reported in the bibliography, the antiproliferative activity of bergenin is evaluated in vitro cellular models, which absolutely do not demonstrate the anticancer activity attributed to the molecule. I therefore ask the authors to make a clear separation between in vitro antiproliferative activity and in vivo studies of the molecule or plant extracts that contain Bergenin. Therefore, change the title of paragraph 5.14 and revise the abstract. Also, make the appropriate distinctions for the other numerous biological activities of the molecule.
Author Response
Reply letter to reviewer 4
We address our greetings and gratitude to reviewer 4 for the rich suggestions, which we accept and correct. In the paper, corrections suggested by reviewer 4 are highlighted i in gray.

Round 2
Reviewer 1 Report
The authors satisfactorily responded to the suggestions.